# Road Network Representation Learning with the Third Law of Geography

Haicang Zhou[1]  Weiming Huang[1]  Yile Chen[1]  Tiantian He[2, 3]  Gao Cong[1]  Yew-Soon Ong[1, 2, 3]

[1] College of Computing and Data Science, Nanyang Technological University, Singapore
[2] Institute of High-Performance Computing, Agency for Science, Technology and Research, Singapore
[3] Centre for Frontier Al Research, Agency for Science, Technology and Research, Singapore
`haicang001@e.ntu.edu.sg, weiming.huang@nateko.lu.se`
`he_tiantian@ihpc.a-star.edu.sg, {yile.chen, gaocong, ASYSOng}@ntu.edu.sg`

## Abstract

Road network representation learning aims to learn compressed and effective vectorized representations for road segments that are applicable to numerous tasks. In this paper, we identify the limitations of existing methods, particularly their overemphasis on the distance effect as outlined in the First Law of Geography. In response, we propose to endow road network representation with the principles of the recent Third Law of Geography. To this end, we propose a novel graph contrastive learning framework that employs geographic configuration-aware graph augmentation and spectral negative sampling, ensuring that road segments with similar geographic configurations yield similar representations, and vice versa, aligning with the principles stated in the Third Law. The framework further fuses the Third Law with the First Law through a dual contrastive learning objective to effectively balance the implications of both laws. We evaluate our framework on two real-world datasets across three downstream tasks. The results show that the integration of the Third Law significantly improves the performance of road segment representations in downstream tasks. Our code is available at `https://github.com/Haicang/Garner`.

## 1 Introduction

Road networks, which form a fundamental infrastructure within urban spaces, describe the geometries and connectivity among road segments in transportation systems. Correspondingly, road networks serve as indispensable components to support numerous smart city applications, such as traffic forecasting [8, 14], route inference [4, 20, 42], and travel time estimation [19]. Motivated by the advancements of graph representation learning [29, 37], the versatile utility of road networks has spurred research into developing effective and expressive representation learning methods for road networks. The objective is to derive functional and easily integrable representations of road segments that can align with the paradigm of neural network models.

Road networks are inherently regarded as graphs, which allow existing methods for road network representation learning to build upon graph representation learning techniques. In particular, apart from encoding the topological information of road networks, these methods further integrate additional spatial characteristics, such as geographical distance [2], which are unique to road networks. In these methods, the inductive bias induced by spatial characteristics is primarily based on the First Law of Geography [25], which states that "*everything is related to everything else, but near things are more related than distant things.*" This principle implies that spatially close road segments tend to have similar representations. For example, skip-gram based methods [3, 39, 40] define the context window based on hops among graph neighbors or spatial distance and derive road segment

representations similar to word2vec [24]. Besides, graph neural networks [11, 17, 36, 43, 52] based methods [2, 15, 44] employ message passing [5, 53] and aggregation among road segments. Both types of methods result in similar representations for connected or proximal road segments [29, 43].

While generally true and applicable, the First Law of Geography does not adequately capture the complexity of urban environments [55], particularly in terms of long-range relationships. Consequently, this limitation compromises the effectiveness of road segment representations in existing methods. This law predominantly emphasizes the distance decay effect, neglecting the influence of semantic factors of different areas on target variables [54]. To mitigate the limitation, the Third Law of Geography [54, 55] was proposed to further consider geographic configurations, stating that "*The more similar geographic configurations of two points (areas), the more similar the values (processes) of the target variable at these two points (areas).*" The term *geographic configuration* refers to the description of spatial neighborhood (or context) around a point (area), and the term *target variable* is the road representation in our context. As a result, two road segments with similar geographic configurations should have similar representations, even if they are disconnected and distant.

Recognizing the potential advantages of integrating the Third Law of Geography, we initiate pioneering research that combines the principles of both the First Law and the Third Law in road network representation learning for the first time. To facilitate this integration, it is essential to leverage data sources that provide comprehensive insights into geographic configurations for road segments. Existing approaches commonly utilize data from OpenStreetMap [27], which includes relatively basic features such as coarse-grained road attributes (location, length, type, etc.). While these data support the condition required in the First Law of Geography and are subsequently processed through specialized model designs, they are insufficient to address the application of the Third Law. To enhance the understanding of geographic configurations, we propose to utilize street view images (SVIs) [7] as an additional data source. Street view images capture the visual context of roads and their surroundings, offering a more nuanced representation of geographic configurations. However, it is still non-trivial to tackle these two principles simultaneously.

First, it is important to effectively enable the integration of the Third Law of Geography within the context of road networks. This law posits that similar representations are expected to be derived for road segments with similar geographic configurations. To this end, the module should capture and reflect the similarity relationships among geographic configurations in the resulting road segment representations, ensuring that road representations faithfully preserve the similarity relationships. Second, it is critical to harmonize the implications of applying both the First and Third Law of Geography in road network representation learning. The First Law emphasizes the importance of spatial proximity, while the Third Law focuses on the significance of similarity in geographic configurations, irrespective of spatial proximity. In real-world scenarios, two distant road segments might exhibit very similar geographic configurations due to similar surrounding buildings and environments. Conversely, two directly connected and proximally close road segments might present vastly different geographic configurations – for example, one adjacent to a commercial area and the other near a park. Given these two conditions, a framework is required to synthesize road segment representations that align with both principles while mitigating potential discrepancies as highlighted.

To resolve these challenges, we propose a new framework, namely **G**eographic **La**w aware **r**oad **ne**twork **r**epresentation learning (`Garner`). First, to effectively enable the integration of the Third Law, we devise a graph contrastive learning framework [22, 37] tailored for road networks. This enhances conventional contrastive learning by incorporating geographic configuration-aware graph augmentation and spectral negative sampling. Specifically, we utilize SVIs to construct a geographic configuration view for road networks, which facilitates the augmentation of edge connections between road segments sharing similar geographic configurations, even if they are geospatially distant. Then, we employ a Simple Graph Convolution (SGC) [43] encoder, which has the property of implicitly reducing the differences between the representations of connected road segments [43], thereby mapping the similarity relationships between geographic configurations to road segment representations in the contrastive learning process. To further align with the Third Law, the proposed contrastive learning framework is equipped with a novel spectral negative sampling technique. This sampling strategy can be mathematically demonstrated to support the principle of the Third Law in a reverse way, ensuring that road segments with dissimilar geographic configurations are represented distinctly. Second, to harmonize the effects of both the First Law and Third Law, we propose a dual contrastive learning objective, which contrasts the topological structure view with the geographic configuration graph view and the spatial proximity graph view. We maintain shared parameters in the SGC encoder

and jointly train the contrastive losses, thus simultaneously learning the consensus and discrepancies of these two laws in a self-supervised manner by minimizing the dual contrastive objective.

Our contributions can be summarized as follows.

- We identify the limitations of existing methods in road network representation learning, and propose the integration of the Third Law of Geography to overcome the shortcomings. To the best of our knowledge, this is the pioneering attempt to integrate the Third Law of Geography in this area.

- We develop a novel graph contrastive learning framework to model the Third Law through geographic configuration-aware graph augmentation and spectral negative sampling. Besides, we balance the influences of two geographic laws via a dual contrastive learning objective.

- We conduct extensive experiments on two real-world road network datasets (i.e., Singapore and New York City), and evaluate our model on three downstream tasks. The experiments demonstrate that the integration of the Third Law significantly enhances the performance of road network representation learning.

## 2 Related work

**Road network representation learning**  These works aim to learn representations for road segments or intersections, for various downstream tasks [4, 8, 14, 20]. Recent studies model a road network as a graph and build their method upon graph representation techniques by including geospatial information, and can be classified into two groups. Some [3, 23, 39, 40] adopt random walks to generate paths and train a skip-gram model [29], incorporating geospatial information based on distance [39] or spatial constraints. Others [2, 15, 44, 48] use graph neural networks [17, 36] to ensure proximal or connected roads have similar representations. Some also include traffic data (e.g., GPS trajectories of vehicles [3, 32]) to enhance the representation. The theoretical support behind these methods is the First Law of Geography [25]. However, this principle has overemphasized spatial proximity and thus [54] proposes the Third Law of Geography, which argues that geographic configurations play critical roles in geospatial data analytics. This paper pioneers research on modeling the geographic configurations and the Third Law for road network representation learning.

**Unsupervised graph representation learning**  Graph representation learning aims to learn representations for graph components like nodes, edges, or entire graphs, where unsupervised node representation learning is most relevant to ours. Despite [12, 13] on masked auto-encoding [10], the majority relies on contrastive learning [35] or its variants, which usually comprises several components: graph augmentation, contrastive strategy, negative sampling, and a loss function (e.g., mutual information (MI) estimator). Graph augmentation [21, 45] is to generate positive [9] or negative [37] samples, though some recent studies try to remove it [26, 49]. Contrastive strategy chooses which two components to contrast, such as node-node contrast [30, 56] or node-graph contrast [9, 21]. Negative sampling is required by the loss. The loss is usually to maximize the MI [9, 37, 50] between an entity with its positive samples and minimize the MI with its negative samples. Several recent studies have also tried to develop new objectives for graphs [26, 47]. Unlike existing studies our method further encodes geographic laws and more data sources to feed the contrastive loss.

## 3 Preliminaries and problem definition

**Definition 1** (Graph). A graph is denoted as $\mathcal{G} = (\mathcal{V}, \mathcal{E}, \boldsymbol{X})$, where $\mathcal{V}$ and $\mathcal{E}$ denote the set of nodes and edges respectively. Let $n = |\mathcal{V}|$ denote the number of nodes and $m = |\mathcal{E}|$ denote the number of edges. $\boldsymbol{X} \in \mathbb{R}^{n \times f'}$ is the feature matrix, with each row $\boldsymbol{X}_i$ representing the features on node $i$. Let $\boldsymbol{A} \in \mathbb{R}^{n \times n}$ denote the adjacency matrix of $\mathcal{G}$, describing the connections in $\mathcal{E}$. If node $i$ and node $j$ are disconnected, then $\boldsymbol{A}_{i,j} = 0$; otherwise $\boldsymbol{A}_{i,j} \neq 0$. By adding self-loops, we define $\tilde{\boldsymbol{A}} = \boldsymbol{A} + \boldsymbol{I}$.

**Definition 2** (Graph Laplacian matrix). The Laplacian matrix of a graph $\mathcal{G}$ is defined as $\boldsymbol{L} := \boldsymbol{D} - \boldsymbol{A}$, where $\boldsymbol{D}$ is the diagonal degree matrix with $\boldsymbol{D}_{i,i}$ as the degree of node $i$ and $\boldsymbol{D}_{i,j} = 0 \quad \forall i \neq j$.

**Definition 3** (Road Network). Road networks are composed of road segments and intersections (junctions) of road segments. We use the term "road" to denote road segment in later sections for brevity. Road networks can be regarded as a graph, which is denoted as $\mathcal{G} = (\mathcal{V}, \mathcal{E}, \boldsymbol{X}, \mathsf{P})$. Each road

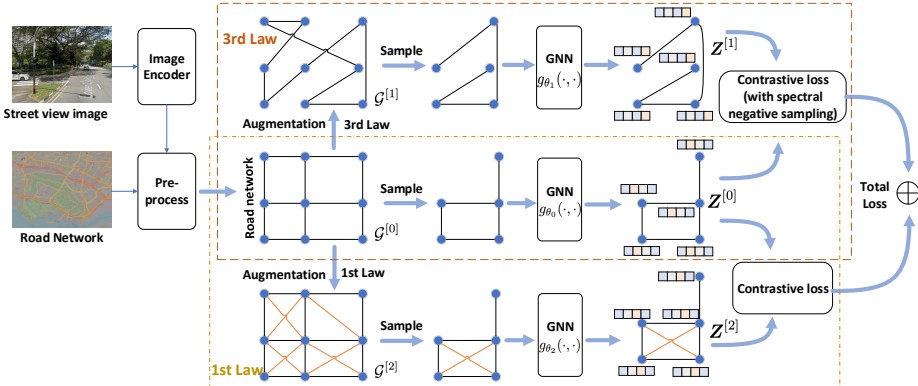

Figure 1: Architecture of Geographic law aware road network representation learning (`Garner`).

segment is modeled as a node, and connected roads are linked by edges. $\boldsymbol{X}_i$ represents the feature vector of road $i$. Besides, road networks contain $\mathbf{P}$, which stores the geospatial locations of nodes.

**Problem Definition** (Road network representation learning). Given road networks $\mathcal{G} = (\mathcal{V}, \mathcal{E}, \boldsymbol{X}, \boldsymbol{C}, \mathbf{P})$, where $\boldsymbol{C}$ denotes the geographic configurations, our objective is to learn a function $\varphi \colon (\boldsymbol{X}, \boldsymbol{C}, \boldsymbol{A}, \mathbf{P}) \to \boldsymbol{Z} \in \mathbb{R}^{n \times f}$, where the $i$-th row $\boldsymbol{Z}_i$ of $\boldsymbol{Z}$ denotes the representation of road $i$.

We defer the definitions and discussions of geographic laws in Appendix A.1.

## 4  Method

In this section, we present the details of `Garner`, which enables the integration of the Third Law of geography for learning road representations: roads with similar geographic configurations should exhibit similar representations, and vice versa. The proposed `Garner` is built on the recent advances in contrastive learning and spectral graph theory. Fig. 1 illustrates the framework of `Garner`, which consists of the following components: (1) Data preprocessing: it produces initial road features that encode the geographic configuration. (2) Graph augmentation: it generates a graph adhering to the Third Law of Geography, where roads with similar geographic configurations are connected. (3) GNN encoder: it serves as the backbone for deriving road representations from both the original graph (the topology of the road network) and the augmented graphs. (4) Graph contrastive loss: enhanced with a spectral negative sampling strategy, it effectively integrates the Third Law (the upper part in Fig. 1). (5) Dual contrastive learning objective: it aims to harmonize the effect of the First Law with the Third Law (lower part in Fig. 1).

`Garner` is inspired by recent studies on contrastive learning [1, 22] showing that by maximizing the mutual information (MI) between different views, the information from these views can be fused properly, and by contrasting different views of the graph the quality of the representation can be improved [9]. By applying a contrastive loss on the augmented and original graph, `Garner` can effectively learn the road representation according to both the third law and the original road network, and fuse their information properly. The cross-contrast strategy also allows `Garner` to fuse the Third Law and the First Law with a proxy, instead of directly contrasting the laws. This allows $\boldsymbol{Z}^{[0]}$ to learn their consensus, while $\boldsymbol{Z}^{[1]}$ and $\boldsymbol{Z}^{[2]}$ can learn the discrepancies of the two laws.

### 4.1  Representation of geographic configuration and data preprocessing

As mentioned in Section 1, we utilize street view images (SVIs) as proxies to represent geographic configurations for road segments. Specifically, each street view image is encoded into a vectorized representation with a pre-trained image encoder (e.g., CLIP [31]). Then, we match these SVIs to roads according to their geospatial locations. As multiple SVIs can correspond to a single road in our datasets, we aggregate the representation of these matched SVIs with average pooling in such cases. After that, the geographic configuration (GC) for road segments can be described as matrix $\boldsymbol{C} \in \mathbb{R}^{n \times c}$. We note that a small portion of roads do not align with any SVIs. For these roads, we set their GC to be the average representation of other roads. Moreover, road segments may possess other

basic attributes available from OpenStreetMap, such as the road type and length. We follow previous literature [2, 3] to encode the features into a matrix $\boldsymbol{X}$. Finally, the GC and additional road features are projected and concatenated to form $\boldsymbol{H}^{(0)} = \text{concat}([\boldsymbol{C}\boldsymbol{W}_c, \boldsymbol{X}\boldsymbol{W}_x])$, which serves as input to our proposed framework.

## 4.2 Geographic configuration aware graph augmentation

We propose a graph augmentation technique according to the similarity of the geographic configuration. We first define a similarity measure for geographic configuration as $\text{sim}(\boldsymbol{C}_i, \boldsymbol{C}_j)$. Here, we use norm-based measure $\text{sim}(\boldsymbol{C}_i, \boldsymbol{C}_j) = 1/(1 + \|\boldsymbol{C}_i - \boldsymbol{C}_j\|)$, while cosine similarity and Gaussian kernel could also be possible choices. Then, we build an augmented similarity graph based on the similarity $\text{sim}(\boldsymbol{C}_i, \boldsymbol{C}_j)$. In a similarity graph, similar node pairs (i.e., node pairs with large similarity scores) will be connected by edges. Popular choices for building the similarity graph include kNN graphs and threshold graphs [38]. We empirically find that the two choices produce very close results, and we therefore choose kNN graph because of their high efficiency. In a kNN graph, each node is connected with k nodes, which have the highest similarity with it, and k is a small number, so the kNN graph is very sparse. We name this process as geographic configuration aware graph augmentation and use matrix $\boldsymbol{S} \in \{0,1\}^{n \times n}$ to denote the adjacency matrix of the augmented similarity graph $\mathcal{G}^{[1]}$.

## 4.3 Graph encoder

To facilitate the integration of the Third Law, we employ Simple Graph Convolution (SGC) [43] as the backbone of our graph encoder to tackle the augmented similarity graph, described as follows:

$$\boldsymbol{Z} = g_{\theta_1}(\boldsymbol{S}, \boldsymbol{H}^{(0)}) = \hat{\boldsymbol{S}}^K \boldsymbol{H}^{(0)} \boldsymbol{\Theta}, \tag{1}$$

where $\hat{\boldsymbol{S}} = \tilde{\boldsymbol{D}}_{\boldsymbol{S}}^{-1/2} \tilde{\boldsymbol{S}} \tilde{\boldsymbol{D}}_{\boldsymbol{S}}^{-1/2}$, $\tilde{\boldsymbol{D}}_{\boldsymbol{S}}$ is the degree matrix of $\tilde{\boldsymbol{S}} := \boldsymbol{S} + \boldsymbol{I}$, $\boldsymbol{\Theta}$ is the learnable parameter, and $\boldsymbol{Z} \in \mathbb{R}^{n \times f}$ is the output representation. As discussed in [43, 56], the simple graph convolutional operation acts as a low pass filter, making the representation $\boldsymbol{Z}$ of connected nodes similar. Specifically, according to [56], the SGC encoder in equation 1 is designed to minimize $\text{tr}(\boldsymbol{Z}^T \boldsymbol{L}_{\boldsymbol{S}} \boldsymbol{Z})$, which is equivalent to the following equation:

$$\text{tr}(\boldsymbol{Z}^T \boldsymbol{L}_{\boldsymbol{S}} \boldsymbol{Z}) = \frac{1}{2} \sum_{i,j} \boldsymbol{S}_{i,j} \|\boldsymbol{Z}_i - \boldsymbol{Z}_j\|^2. \tag{2}$$

The details of the calculation can be found in Appendix B.1. In this equation, $\boldsymbol{S}_{i,j}$ is non-zero if and only if node $i$ and $j$ are connected in the augmented graph $\mathcal{G}^{[1]}$. Therefore, by minimizing $\text{tr}(\boldsymbol{Z}^T \boldsymbol{L}_{\boldsymbol{S}} \boldsymbol{Z})$, the difference between $\boldsymbol{Z}_i$ and $\boldsymbol{Z}_j$ is reduced for every connected node pair, thus aligning with the objective of the Third Law, where the representations of nodes (i.e., road segments) with similar geographic configuration are minimized. In addition, another SGC encoder with different learnable parameters is used to encode the nodes in the original graph $\mathcal{G}^{[0]}$, which represents the topological structure of road networks. As a result, we obtain two outputs, $\boldsymbol{Z}^{[0]}$ from the original graph $\mathcal{G}^{[0]}$, and $\boldsymbol{Z}^{[1]}$ from the augmented graph $\mathcal{G}^{[1]}$. Given the large number of road segments in a city, we adopt a sub-sampling technique to enhance the efficiency and scalability. In particular, in each iteration in training, we maintain the same set of nodes for each graph view, and only edges connecting sampled nodes are retained. The subgraphs are then processed through the graph encoders.

## 4.4 Contrastive loss

Previous studies show that information from different views can be fused properly by maximizing their mutual information (MI) [1], which can also improve the quality of representations [9]. Inspired by these, we maximize the MI between the original graph $\mathcal{G}^{[0]}$ and the augmented graph $\mathcal{G}^{[1]}$,

$$\mathcal{L}_1 = -\frac{1}{|\mathcal{V}|} \sum_{i=1}^{|\mathcal{V}|} \left\{ \text{MI}(\boldsymbol{Z}_i^{[0]}, \boldsymbol{Z}_g^{[1]}) + \text{MI}(\boldsymbol{Z}_i^{[1]}, \boldsymbol{Z}_g^{[0]}) \right\}. \tag{3}$$

Here $|\mathcal{V}|$ denotes the number of nodes in the graph. $\boldsymbol{Z}_g^{[0]} \in \mathbb{R}^f$ is the representation of the whole graph by applying a graph pooling operation on $\boldsymbol{Z}^{[0]}$. In our implementation, we choose the mean

pooling. The MI estimator $\text{MI}(\cdot, \cdot)$ is a widely used one based on Jensen-Shannon divergence [37].

$$\text{MI}(\boldsymbol{Z}_i^{[0]}, \boldsymbol{Z}_g^{[1]}) = \mathbb{E}_{(\mathcal{G}^{[0]}, \mathcal{G}^{[1]})} \left[ \log \mathcal{D}(\boldsymbol{Z}_i^{[0]}, \boldsymbol{Z}_g^{[1]}) \right] + \mathbb{E}_{(\bar{\mathcal{G}}^{[0]}, \mathcal{G}^{[1]})} \left[ \log(1 - \mathcal{D}(\bar{\boldsymbol{Z}}_i^{[0]}, \boldsymbol{Z}_g^{[1]})) \right], \quad (4)$$

$$\text{MI}(\boldsymbol{Z}_i^{[1]}, \boldsymbol{Z}_g^{[0]}) = \mathbb{E}_{(\mathcal{G}^{[1]}, \mathcal{G}^{[0]})} \left[ \log \mathcal{D}(\boldsymbol{Z}_i^{[1]}, \boldsymbol{Z}_g^{[0]}) \right] + \mathbb{E}_{(\bar{\mathcal{G}}^{[1]}, \mathcal{G}^{[0]})} \left[ \log(1 - \mathcal{D}(\bar{\boldsymbol{Z}}_i^{[1]}, \boldsymbol{Z}_g^{[0]})) \right]. \quad (5)$$

where the discriminator $\mathcal{D}$ is achieved by a bilinear layer (i.e., $\mathcal{D}(\boldsymbol{a}, \boldsymbol{b}) = \boldsymbol{a}^T \boldsymbol{W} \boldsymbol{b}$ ) [22]. $\bar{\mathcal{G}}^{[0]}$ and $\bar{\mathcal{G}}^{[1]}$ are negative samples required by the mutual information estimator. Specifically, $\bar{\mathcal{G}}^{[0]}$ is the negative sample generated by shuffling the rows of inputs $\boldsymbol{H}^{(0)}$, following [22, 37], and then $\bar{\boldsymbol{Z}}^{[0]}$ is produced through the graph encoder $g_{\theta_0}$. We then explain how to generate $\bar{\mathcal{G}}^{[1]}$.

### 4.5 Spectral negative sampling

In graph contrastive learning, negative sampling is usually implemented by graph corruption such as feature shuffling [37] and edge modification [46]. In the road network context, we extend beyond these conventional methods by introducing a novel spectral negative sampling technique to integrate the Third Law of Geography. To be specific, the Third Law not only posits that "roads with similar geographic configurations should have similar representations," but also implies that "roads with dissimilar geographic configurations should have dissimilar representations." Our proposed strategy elegantly refines the objective in Equation 2 in the contrastive learning process, thereby implicitly addressing the reverse implication of the Third Law. [1]

To achieve this, we recall that equation 2 relates closely to the objective of the sparsest cut problem (Chapter 10 of [34] and [49]), which seeks to find cuts that minimize the number of edges between subsets of nodes. Correspondingly, nodes are densely connected within each subset, while between different subsets, nodes are sparsely connected.

$$usc_{\mathcal{G}} := \min_S \frac{\mathcal{E}(S, \mathcal{V} - S)}{|S||\mathcal{V} - S|}, \quad (6)$$

where the numerator denotes the number of edges across node sets $S$ and $\mathcal{V} - S$, and the denominator is the multiplication of number of nodes in the two sets. $usc_{\mathcal{G}}$ can be computed as

$$usc_{\mathcal{G}} = \min_{\boldsymbol{x} \in \{0,1\}^n - \{\boldsymbol{0}, \boldsymbol{1}\}} \frac{\sum_{(i,j) \in \mathcal{E}} (\boldsymbol{x}_i - \boldsymbol{x}_j)^2}{\sum_{(i,j)} (\boldsymbol{x}_i - \boldsymbol{x}_j)^2} = \min_{\boldsymbol{x} \in \{0,1\}^n - \{\boldsymbol{0}, \boldsymbol{1}\}} \frac{\boldsymbol{x}^T \boldsymbol{L}_{\mathcal{G}} \boldsymbol{x}}{\boldsymbol{x}^T \boldsymbol{L}_{\mathcal{K}} \boldsymbol{x}}, \quad (7)$$

where $\boldsymbol{x}_i$ is the $i$-th element of vector $\boldsymbol{x}$, and $\mathcal{K}$ is a complete graph, which has the same nodes as $\mathcal{G}$. Following [49], we apply a continuous relaxation in this formula and extend it to the matrix form:

$$\min_{\boldsymbol{Z} \in \mathbb{R}^{n \times f}} \frac{\text{tr}(\boldsymbol{Z}^T \boldsymbol{L_S} \boldsymbol{Z})}{\text{tr}(\boldsymbol{Z}^T \boldsymbol{L_K} \boldsymbol{Z})}, \quad (8)$$

where $\boldsymbol{L_S}$ is the graph Laplacian matrix by regarding $\boldsymbol{S}$ as the adjacency matrix, and $\boldsymbol{L_K}$ is the graph Laplacian matrix of the complete graph $\mathcal{K}$. Minimizing equation 8 is equivalent to minimizing the numerator and meanwhile maximizing the denominator, with the same $\bar{\boldsymbol{Z}}$. Recall that the SGC encoder (equation 1) has the effect of minimizing the numerator $\text{tr}(\boldsymbol{Z}^T \boldsymbol{L_S} \boldsymbol{Z})$. Subsequently, we design the *negative* sample based on $\boldsymbol{Z}$ and $\mathcal{K}$. The negative sample can achieve that by discriminating positive samples from negatives, the model has the effect of maximizing the denominator $\text{tr}(\boldsymbol{Z}^T \boldsymbol{L_K} \boldsymbol{Z})$. This can be achieved by another SGC with the same parameter as in equation 1:

$$\bar{\boldsymbol{Z}} = g_{\theta_1}(\hat{\boldsymbol{A}}_{\mathcal{K}}, \boldsymbol{H}^{(0)}) = \hat{\boldsymbol{A}}_{\mathcal{K}}^{K'} \boldsymbol{Z} = \hat{\boldsymbol{A}}_{\mathcal{K}}^{K'} \hat{\boldsymbol{S}}^K \boldsymbol{H}^{(0)} \boldsymbol{\Theta} = \hat{\boldsymbol{A}}_{\mathcal{K}} \boldsymbol{H}^{(0)} \boldsymbol{\Theta}, \quad (9)$$

where $\boldsymbol{A}_{\mathcal{K}}$ is the adjacency matrix of $\mathcal{K}$, $\hat{\boldsymbol{A}}_{\mathcal{K}} = \boldsymbol{D}_{\boldsymbol{A}_{\mathcal{K}}}^{-0.5} \boldsymbol{A}_{\mathcal{K}} \boldsymbol{D}_{\boldsymbol{A}_{\mathcal{K}}}^{-0.5} = \boldsymbol{A}_{\mathcal{K}}/n$. $\bar{\boldsymbol{Z}} = g_{\theta_1}(\hat{\boldsymbol{A}}_{\mathcal{K}}, \boldsymbol{H}^{(0)})$ achieves minimizing the denominator. (See the details of the computation in Appendix B.2.) Finally, regarding $\mathcal{K}$ (with $\bar{\boldsymbol{Z}}$ as its output) as the negative sample and discriminating positive samples from it in the MI estimator (equation 4 & 5) achieve maximizing the denominator $\text{tr}(\boldsymbol{Z}^T \boldsymbol{L_K} \boldsymbol{Z})$ in equation 8.

However, performing SGC on the complete graph $\mathcal{K}$ entails a time and space complexity of $O(n^2)$, which is computationally infeasible for large graphs. To tackle this, we further conduct an efficient approximation for the complete graph $\mathcal{K}$ according to spectral graph sparsification [33].

$$(1 - \frac{2\sqrt{d-1}}{d}) \, \text{tr}(\boldsymbol{Z}^T \boldsymbol{L}_{\widetilde{\mathcal{K}}} \boldsymbol{Z}) \leq \text{tr}(\boldsymbol{Z}^T \boldsymbol{L}_{\mathcal{K}} \boldsymbol{Z}) \leq (1 + \frac{2\sqrt{d-1}}{d}) \, \text{tr}(\boldsymbol{Z}^T \boldsymbol{L}_{\widetilde{\mathcal{K}}} \boldsymbol{Z}), \quad (10)$$

---

[1] For notation simplicity, we omit superscripts and use $\boldsymbol{Z}$ to denote $\boldsymbol{Z}^{[1]}$ in this section.

where $\tilde{\mathcal{K}}$ is a d-regular graph (i.e., each node has d edges connected to it) with "all of whose non-zero Laplacian eigenvalues lie between $d - 2\sqrt{d-1}$ and $d + 2\sqrt{d-1}$" and each edge weight as $n/d$. [2] Consequently, we can optimize $\mathrm{tr}(\boldsymbol{Z}^T \boldsymbol{L}_{\mathcal{K}} \boldsymbol{Z})$ by optimizing $\mathrm{tr}(\boldsymbol{Z}^T \boldsymbol{L}_{\tilde{\mathcal{K}}} \boldsymbol{Z})$. Then we simply replace $\hat{\boldsymbol{A}}_{\mathcal{K}}$ in the right hand side of equation 9 with $\hat{\boldsymbol{A}}_{\tilde{\mathcal{K}}}$ and get the outputs of the negative sample as

$$\bar{\boldsymbol{Z}} = \hat{\boldsymbol{A}}_{\tilde{\mathcal{K}}} \boldsymbol{H}^{(0)} \boldsymbol{\Theta}, \tag{11}$$

which works well in our experiments.

To summarize, we generate the negative sample $\bar{\mathcal{G}}^{[1]}$ as a d-regular graph $\tilde{\mathcal{K}}$ with the same node set $\mathcal{V}$, with the same node feature as $\mathcal{G}^{[1]}$. Then we perform SGC on the negative sample, and finally use the output representation $\bar{\boldsymbol{Z}}$ to compute mutual information in 5. The negative sampling strategy is inspired by the sparest cut and can maximize $\mathrm{tr}(\boldsymbol{Z}^T \boldsymbol{L}_{\mathcal{K}} \boldsymbol{Z}) = 0.5 \sum_{i=1}^{n} \sum_{j=1}^{n} \|\boldsymbol{Z}_i - \bar{\boldsymbol{Z}}_j\|^2$, where the majority node pairs $(i, j)$ have dissimilar geographic configuration, because of the sparsity of the positive sample $\mathcal{G}^{[1]}$ [34]. Therefore, we can achieve the reverse implication of the Third Law – roads with dissimilar geographic configuration have dissimilar representations.

### 4.6 Fusing the Third Law and the First Law

While the integration of the Third Law has been effectively achieved through our module designs, the First Law remains beneficial, especially in regions that manifest identical functionality, thus encouraging representations of nearby roads within the regions to be similar. Therefore, it is important to further incorporate the inductive bias introduced by this law into the contrastive learning process. To achieve this, we adopt another graph augmentation technique based on graph diffusion [9], which generates another graph view $\mathcal{G}^{[2]}$ with the following adjacency matrix:

$$\boldsymbol{P} = \alpha(\boldsymbol{I} - (1 - \alpha)\tilde{\boldsymbol{D}}^{-1/2} \boldsymbol{A} \tilde{\boldsymbol{D}}^{-1/2})^{-1}, \tag{12}$$

which can be fast approximated by [18]. Then the SGC encoder $g_{\theta_2}$ is employed to produce the output $\boldsymbol{Z}^{[2]}$ from $\mathcal{G}^{[2]}$. The graph diffusion process connects near roads, and the SGC encoder can pull the near roads to have similar representation due to the property of SGC (as in section 4.3). Then we use the same loss for $\mathcal{G}^{[0]}$ and the augmented graph $\mathcal{G}^{[2]}$.

$$\mathcal{L}_2 = -\frac{1}{|\mathcal{V}|} \sum_{i=1}^{|\mathcal{V}|} \left\{ \mathrm{MI}(\boldsymbol{Z}_i^{[0]}, \boldsymbol{Z}_g^{[2]}) + \mathrm{MI}(\boldsymbol{Z}_i^{[2]}, \boldsymbol{Z}_g^{[0]}) \right\} \tag{13}$$

By introducing this component, our `Garner` is endowed with a dual contrastive objective that maximizes the mutual information between the topological structure with the geographic configuration view, as well as the spatial proximity view, in alignment with the principles of these two laws. Then we train the model (Fig. 1) by combining $\mathcal{L}_1$ and $\mathcal{L}_2$:

$$\mathcal{L} = \mathcal{L}_1 + \mathcal{L}_2. \tag{14}$$

We find that the model adeptly learns to balance these considerations through parameter sharing in the SGC encoder ($g_{\theta_0}$), performing well without the need for manual tuning the weights of the two loss functions. Therefore, we do not introduce additional hyper-parameters to adjust their weights. During inference, we aggregate the outputs from the three graph encoders as the road segment representations for downstream tasks: $\boldsymbol{Z} = (\boldsymbol{Z}^{[0]} + \boldsymbol{Z}^{[1]} + \boldsymbol{Z}^{[2]})/3$.

## 5 Experiments

In this section, we evaluate the proposed method and the output road representation following previous literature [2, 3]. The road representation is evaluated on three downstream tasks. We also perform a case study to show the impact of the Third Law, and ablation studies and hyper-parameter sensitivity tests to analyze the proposed method.

### 5.1 Experimental setups

---

[2]For implementation, the adjacency matrix of $\tilde{K}$ will be normalized to its degree matrix, and thus the weight $n/d$ will not change the scale of $\boldsymbol{Z}$.

**Datasets**  We use data from two cities, i.e. *Singapore* and *New York City* (*NYC*). The datasets include road networks from OpenStreetMap (OSM, [27]) and street view images (SVIs) from Google Map ([7]). The statistics can be found in Table 1.

Table 1: Dataset Statistics

| City | # Roads | # Edges | # SVIs |
|------|---------|---------|--------|
| Singapore | 45,243 | 138,843 | 136,399 |
| NYC | 139,320 | 524,565 | 254,239 |

**Downstream tasks**  The road representation is evaluated on three downstream tasks: *road function prediction*, *road traffic inference*, and *visual road retrieval*. Road function prediction is a classification task to determine the functionality of a road. Road traffic inference is a regression task predicting the average speed of vehicles on each road. Visual road retrieval involves finding the roads where the road image should be located. While road traffic inference is widely used in previous literature [2, 3, 23], we introduce road function prediction and visual road retrieval as two new but meaningful evaluation tasks for road representations. More details for downstream tasks can be found in Appendix C.1.

**Evaluation metrics**  For road function prediction, we use Micro-F1, Macro-F1, and AUROC (the area under the ROC curve) as the evaluation metrics. For road traffic inference, we use MAE (mean absolute error), RMSE (root mean square error), and MAPE (mean absolute percentage error) as the evaluation metrics. For visual road retrieval, we use recall@10 and MRR (mean reciprocal rank).

**Baselines**  The proposed `Garner` is compared with seven strong baselines, including Deepwalk [29], MVGRL [9], CCA-SSG [47], GGD [51], RFN [15], SRN2Vec [39] and SARN [2]. Some other recent methods [3, 23, 32, 44] include other data types (e.g., GPS trajectory data of vehicles) as inputs. However, GPS trajectory data are only available in very few cities, and we did not find them for one of our datasets (NYC). Thus, we cannot run these methods for comparison in our experiments. More details of the baselines can be found in Appendix C.2.

**Hyper-parameter settings**  We use the same hyper-parameters on all the datasets. The road features, and image embeddings are projected into 256 dimensions. The $k = 6$ in kNN graph for geographic configuration aware graph augmentation, and $d = 22$ for spectral negative sampling. The hyper-parameters for graph diffusion is $\alpha = 0.2$, as suggested by [18]. The hidden dimension and the dimension of the representation are set as 512. Other detailed settings can be found in Appendix C.3.

## 5.2  Experimental results

Table 2: Results in Road Function Prediction, with the best in **bold** and the second best underlined

| Methods | Singapore | | | NYC | | |
|---------|-----------|---|---|-----|---|---|
| | Micro-F1 (%) ↑ | Macro-F1 (%) ↑ | AUROC (%) ↑ | Micro-F1 (%) ↑ | Macro-F1 (%) ↑ | AUROC (%) ↑ |
| Deepwalk | 62.76 ± 0.49 | 13.30 ± 0.10 | 63.23 ± 0.47 | 78.09 ± 0.18 | 14.62 ± 0.02 | 58.49 ± 0.33 |
| MVGRL | 66.61 ± 0.50 | 30.67 ± 0.66 | 74.34 ± 0.46 | 78.23 ± 0.23 | 17.39 ± 0.23 | 69.96 ± 0.35 |
| CCA-SSG | 64.28 ± 0.37 | 22.55 ± 0.49 | 70.26 ± 0.37 | 78.20 ± 0.24 | 15.97 ± 0.15 | 68.15 ± 0.24 |
| GGD | 64.21 ± 0.39 | 20.58 ± 0.40 | 68.97 ± 0.40 | 78.14 ± 0.25 | 15.75 ± 0.16 | 66.11 ± 0.33 |
| RFN | 62.75 ± 0.44 | 12.85 ± 0.06 | 54.64 ± 0.44 | oom | oom | oom |
| SRN2Vec | 64.02 ± 0.45 | 22.47 ± 0.37 | 71.18 ± 0.40 | oom | oom | oom |
| SARN | 66.49 ± 0.47 | 22.59 ± 0.51 | 72.74 ± 0.50 | 78.14 ± 0.21 | 14.62 ± 0.02 | 68.54 ± 0.30 |
| Garner | **81.40** ± 0.30 | **62.45** ± 0.64 | **93.27** ± 0.22 | **82.97** ± 0.16 | **47.22** ± 0.42 | **89.30** ± 0.21 |

*"oom" means out-of-memory.*

Table 3: Results in Road Traffic Inference, with the best in **bold** and the second best underlined

| Methods | Singapore | | | NYC | | |
|---------|-----------|---|---|-----|---|---|
| | MAE ↓ | RMSE ↓ | MAPE ↓ | MAE ↓ | RMSE ↓ | MAPE ↓ |
| Deepwalk | 3.43 ± 0.03 | 4.31 ± 0.05 | 0.721 ± 0.038 | 4.31 ± 0.03 | 5.92 ± 0.05 | 0.267 ± 0.002 |
| MVGRL | 3.04 ± 0.04 | 3.82 ± 0.04 | 0.629 ± 0.041 | 3.91 ± 0.02 | 5.16 ± 0.03 | 0.243 ± 0.001 |
| CCA-SSG | 3.31 ± 0.03 | 4.15 ± 0.04 | 0.674 ± 0.037 | 4.03 ± 0.03 | 5.34 ± 0.04 | 0.253 ± 0.003 |
| GGD | 3.37 ± 0.03 | 4.27 ± 0.04 | 0.684 ± 0.039 | 4.80 ± 0.03 | 6.63 ± 0.06 | 0.267 ± 0.002 |
| RFN | 3.54 ± 0.03 | 4.48 ± 0.04 | 0.717 ± 0.046 | oom | oom | oom |
| SRN2Vec | 3.44 ± 0.04 | 4.47 ± 0.05 | 0.569 ± 0.025 | oom | oom | oom |
| SARN | 3.40 ± 0.03 | 4.32 ± 0.05 | 0.697 ± 0.038 | 4.66 ± 0.04 | 6.39 ± 0.07 | 0.262 ± 0.002 |
| Garner | **2.80** ± 0.03 | **3.52** ± 0.04 | **0.579** ± 0.030 | **3.30** ± 0.02 | **4.40** ± 0.03 | **0.207** ± 0.002 |

*"oom" means out-of-memory.*

We compare the proposed `Garner` with baselines in three downstream tasks, and the experimental results can be found in Table 2, 3 and 4. In road function prediction and road traffic inference, we report the mean results and standard deviation of 30 runs for each model. Among all the tasks, `Garner` performs significantly better than all the baselines. Specifically, in road function prediction, `Garner` outperforms the best baseline by up to 22% in Micro-F1, 171% in Macro-F1, and 25% in AUROC. In road traffic inference tasks, `Garner` outperforms the best baselines by up to 18.5% in MAE, 17% in RMSE,

Table 4: Results on Visual Road Retrieval, with the best in **bold** and the second best underlined

| Methods | Singapore | | NYC | |
|---|---|---|---|---|
| | Recall@10 ↑ | MRR ↑ | Recall@10 ↑ | MRR ↑ |
| Deepwalk | 0.0083 | 0.0913 | 0.0013 | 0.0709 |
| MVGRL | 0.0088 | 0.0818 | 0.0324 | 0.1071 |
| CCA-SSG | 0.0112 | 0.0755 | 0.0036 | 0.0807 |
| GGD | 0.0095 | 0.0920 | 0.0019 | 0.0695 |
| RFN | 0.0030 | 0.0766 | oom | oom |
| SRN2Vec | 0.0123 | 0.0725 | oom | oom |
| SARN | 0.0143 | 0.1019 | 0.0036 | 0.0766 |
| Garner | **0.4600** | **0.3387** | **0.5531** | **0.2985** |

*"oom" means out-of-memory.*

and 17.4% in MAPE. This is because the geographic configuration can provide more details about the roads. For example, the geographic configurations of roads in living apartments and business regions are very different. Thus, the functions of these two roads can be more easily discriminated according to geographic configuration aware road representation. As for road traffic inference, the geographic configuration provides more details about the conditions of roads. Thus, it is beneficial for traffic systems. On visual road retrieval, all the baselines give results similar to random guesses, while `Garner` gives decent results. The results show that the street view images and geographic configurations provide very different information, which is not presented in road network data. But that information can be well captured by our proposed `Garner`. We also find that some baselines use up GPU memory and cannot run on Tesla GPUs. We discuss the scalability issue in Appendix C.4.

## 5.3 Model analysis

**Case study** We conduct a case study comparing the First and Third Laws of Geography. The case study will show the distinctive effect of the Third Law. We randomly select an anchor road, compute its representation similarity with other roads (according to cosine similarity), and display the top 10 most similar roads. The anchor road is in red, the top 10 similar roads by the First Law are shown in blue, and those by both laws are in orange. The results can be found in Fig. 2. With only the First Law, similar roads are much closer to the anchor. With both laws, similar roads are further but have similar geographic configurations, as shown by comparing street view images. This demonstrates that the Third Law ensures similar representations for roads with similar configurations, regardless of distance.

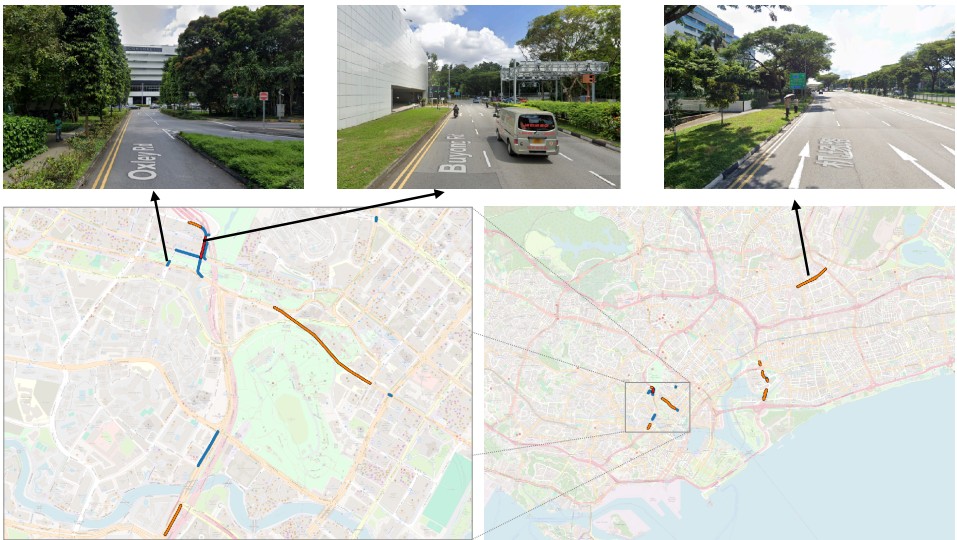

Figure 2: Case study: the top 10 most similar roads found by the First Law only and both laws.

**Ablation studies**   We conduct ablation studies by: (1) gradually removing the components in `Garner`; (2) using different similarity measure to build the augmented graph. The ablation results on road function predictions are listed in Table 5 and 6, while results on other tasks can be found in Appendix C.5. We show that the street view images, utilized to describe the geographic configurations, provide significant improvements in the quality of the road representation by comparing "`Garner` - sns - aug" and "`Garner` - sns - aug - SVI". We also find that explicitly modeling the Third Law of geography provides significant improvements by comparing "`Garner`" and "`Garner` - sns - aug". Spectral negative sampling also provides obvious improvements in the Macro-F1 score and visual road retrieval. As for the choice of similarity measure, kNN similarity graphs are better than threshold similarity graphs on Road Function prediction, while slightly worse on Road Traffic Inference. Meanwhile using euclidean distance generate similar results as using cosine similarity. As building kNN graphs is much faster than threshold graphs, we choose kNN graphs as the default setting.

Table 5: Ablation studies on Road Function Prediction

| Methods | Singapore | | | NYC | | |
|---|---|---|---|---|---|---|
| | Micro-F1 (%) ↑ | Macro-F1 (%) ↑ | AUROC (%) ↑ | Micro-F1 (%) ↑ | Macro-F1 (%) ↑ | AUROC (%) ↑ |
| `Garner` - sns - aug - SVI | $66.61 \pm 0.50$ | $30.67 \pm 0.66$ | $74.34 \pm 0.46$ | $78.23 \pm 0.23$ | $17.39 \pm 0.23$ | $69.96 \pm 0.35$ |
| `Garner` - sns - aug | $74.78 \pm 0.32$ | $50.21 \pm 0.60$ | $88.21 \pm 0.30$ | $80.64 \pm 0.22$ | $37.14 \pm 0.44$ | $85.30 \pm 0.25$ |
| `Garner` - sns | $80.65 \pm 0.31$ | $60.57 \pm 0.68$ | $92.46 \pm 0.23$ | $82.62 \pm 0.19$ | $45.78 \pm 0.52$ | $88.61 \pm 0.18$ |
| `Garner` | $81.40 \pm 0.30$ | $62.45 \pm 0.64$ | $93.27 \pm 0.22$ | $82.97 \pm 0.16$ | $47.22 \pm 0.42$ | $89.30 \pm 0.21$ |

*"- sns" means to generate negative samples only with feature shuffling. "- aug" means without geographic configuration aware graph augmentation. "- SVI" means without street view images as inputs.*

Table 6: Ablation studies of similarity measures on Road Function Prediction

| Methods | Singapore | | | NYC | | |
|---|---|---|---|---|---|---|
| | Micro-F1 (%) ↑ | Macro-F1 (%) ↑ | AUROC (%) ↑ | Micro-F1 (%) ↑ | Macro-F1 (%) ↑ | AUROC (%) ↑ |
| knn-dist | $81.40 \pm 0.30$ | $62.45 \pm 0.64$ | $93.27 \pm 0.22$ | $82.97 \pm 0.16$ | $47.22 \pm 0.42$ | $89.30 \pm 0.21$ |
| knn-cos | $81.30 \pm 0.34$ | $62.26 \pm 0.56$ | $92.94 \pm 0.21$ | $82.95 \pm 0.18$ | $46.98 \pm 0.56$ | $89.13 \pm 0.18$ |
| threshold-dist | $79.01 \pm 0.56$ | $56.71 \pm 0.93$ | $91.15 \pm 0.26$ | $81.39 \pm 0.23$ | $40.16 \pm 0.50$ | $86.32 \pm 0.23$ |
| threshold-cos | $79.45 \pm 0.32$ | $57.76 \pm 0.53$ | $91.71 \pm 0.23$ | $82.51 \pm 0.20$ | $45.07 \pm 0.59$ | $88.73 \pm 0.19$ |

**Parameter sensitivity analysis**   We conduct sensitivity analysis on two hyper-parameters introduced by our method. They are the degree ($k$) of the augmented kNN similarity graph and the degree ($d$) of the negative graph. The results are visualized in Fig. 3, and more results are listed in Appendix C.6. In these figures, the shadows show the standard deviation. In our experiments, we find that the results are not sensitive to the hyper-parameters.

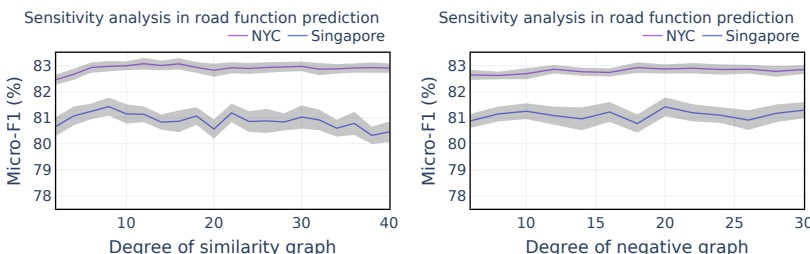

Figure 3: Sensitivity analysis of hyper-parameter $k$ and $d$.

## 6   Conclusion

In this paper, we embark on pioneering research investigating the *Third Law of geography* for road network representation learning. To model the Third Law, we introduce street view images to capture the geographic configuration and design a new contrastive learning framework with geographic configuration aware graph augmentation and spectral negative sampling. The experiments show that the proposed method brings significant improvements in road network representation and downstream tasks. There are certainly many future directions, such as modeling more geographic laws, designing new evaluations, and using other kinds of real-world data.

## Acknowledgments and Disclosure of Funding

This research was supported in part by Distributed Smart Value Chain programme which is funded under the Singapore RIE2025 Manufacturing, Trade and Connectivity (MTC) Industry Alignment Fund-Pre-Positioning (Award No: M23L4a0001), as well as cash and in-kind contribution from Singapore Telecommunications Limited (Singtel), through Singtel Cognitive and Artificial Intelligence Lab for Enterprises (SCALE@NTU)."

This research was also partially funded by the Knut and Alice Wallenberg Foundation (KAW 2019.0550).

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

## A  Backgrounds

### A.1  Discussion on the Laws of Geography

**Definition 4** (The First Law of Geography, or Tobler's First Law of Geography, [25]). Everything is related to everything else, but near things are more related than distant things.

**Definition 5** (The Third Law of Geography, [54]). The more similar geographic configurations of two points (areas), the more similar the values (processes) of the target variable at these two points (areas).

To illustrate the Third Law of Geography, consider two roads surrounded by residential buildings. Even though two roads are located very far apart, according to the third law, as they have very similar geographic configurations, they should still have similar representations.

As extensively discussed in the paper, this study has been substantially informed by the theories in geography and geographic information science, particularly the First Law of Geography and the Third Law of Geography. In fact, our study has also been partially informed by the Second Law of Geography, which is arguably about spatial heterogeneity. The Second Law of Geography implies that geographic variables and processes exhibit uncontrolled variance [6]. We are informed by this law from the perspective that the spatial proximity graph and the original graph only connect those close enough road segments, rather than forming complete graphs. This methodological choice acknowledges the considerable heterogeneity in the influence exerted by more distant roads.

## B  Additional calculation and proofs

### B.1  Details of calculation in section 4.2

$$
\begin{aligned}
\mathrm{tr}(\boldsymbol{Z}^T \boldsymbol{L_S} \boldsymbol{Z}) &= \sum_k \sum_{i,j} \boldsymbol{Z}_{i,k} \boldsymbol{Z}_{j,k} (\boldsymbol{L_S})_{i,j} \\
&= \sum_k \left( \sum_i \boldsymbol{Z}_{i,k}^2 (\boldsymbol{L_S})_{i,i} + \sum_{i \neq j} \boldsymbol{Z}_{i,k} \boldsymbol{Z}_{j,k} (\boldsymbol{L_S})_{i,j} \right) \\
&= \frac{1}{2} \sum_k \left( \sum_i \boldsymbol{Z}_{i,k}^2 (\boldsymbol{L_S})_{i,i} + \sum_{i \neq j} 2 \boldsymbol{Z}_{i,k} \boldsymbol{Z}_{j,k} (\boldsymbol{L_S})_{i,j} + \sum_j \boldsymbol{Z}_{j,k}^2 (\boldsymbol{L_S})_{j,j} \right) \\
&= \frac{1}{2} \sum_k \left( \sum_{i \neq j} \boldsymbol{Z}_{i,k}^2 (\boldsymbol{S}_{i,j} + 2 \boldsymbol{Z}_{i,k} \boldsymbol{Z}_{j,k}(-\boldsymbol{S}_{i,j}) + \boldsymbol{Z}_{j,k}^2 \boldsymbol{S}_{i,j}) \right) \\
&= \frac{1}{2} \sum_k \sum_{i \neq j} \boldsymbol{S}_{i,j} (\boldsymbol{Z}_{i,k} - \boldsymbol{Z}_{j,k})^2 \\
&= \frac{1}{2} \sum_{i,j} \boldsymbol{S}_{i,j} \| \boldsymbol{Z}_i - \boldsymbol{Z}_j \|^2 .
\end{aligned}
$$

### B.2  Details of calculation in section 4.5

Here we show that $\hat{\boldsymbol{A}}_{\mathcal{K}}^{K'} \hat{\boldsymbol{S}}^K \boldsymbol{H}^{(0)} \boldsymbol{\Theta} = \hat{\boldsymbol{A}}_{\mathcal{K}} \boldsymbol{H}^{(0)} \boldsymbol{\Theta}$.

Recall that $\hat{\boldsymbol{S}} = \tilde{\boldsymbol{D}}_{\boldsymbol{S}}^{-1/2} \tilde{\boldsymbol{S}} \tilde{\boldsymbol{D}}_{\boldsymbol{S}}^{-1/2}$ is a symmetric matrix where each row or column is summed to 1. $\boldsymbol{A}_{\mathcal{K}}$ is a matrix with $\boldsymbol{A}_{i,j} = 1 \ \forall i, j$. $\hat{\boldsymbol{A}}_{\mathcal{K}} = \boldsymbol{A}_{\mathcal{K}}/n$. Thus

$$
(\hat{\boldsymbol{A}}_{\mathcal{K}} \hat{\boldsymbol{S}})_{i,j} = \sum_l \frac{1}{n} \times \hat{\boldsymbol{S}}_{l,j} = \frac{1}{n} \sum_l \hat{\boldsymbol{S}}_{l,j} = \frac{1}{n} .
$$

Then we have

$$
\hat{\boldsymbol{A}}_{\mathcal{K}} \hat{\boldsymbol{S}} = \hat{\boldsymbol{A}}_{\mathcal{K}} . \tag{15}
$$

For the power of $\hat{\boldsymbol{A}}_{\mathcal{K}}$, we have the following result for every $i$ and $j$

$$
(\hat{\boldsymbol{A}}_{\mathcal{K}} \hat{\boldsymbol{A}}_{\mathcal{K}})_{i,j} = \sum_l (\hat{\boldsymbol{A}}_{\mathcal{K}})_{i,l} (\hat{\boldsymbol{A}}_{\mathcal{K}})_{l,j} = \sum_l \frac{1}{n} \times \frac{1}{n} = \frac{1}{n} ,
$$

where $n$ is the dimension of $\hat{\boldsymbol{A}}_{\mathcal{K}}$ (i.e., $\hat{\boldsymbol{A}}_{\mathcal{K}} \in \mathbb{R}^{n \times n}$). The matrix form of the result is

$$\hat{\boldsymbol{A}}_{\mathcal{K}}\hat{\boldsymbol{A}}_{\mathcal{K}} = \hat{\boldsymbol{A}}_{\mathcal{K}} \tag{16}$$

Combining them together, we have

$$\begin{aligned}
\hat{\boldsymbol{A}}_{\mathcal{K}}^{K'}\hat{\boldsymbol{S}}^K \boldsymbol{H}^{(0)}\boldsymbol{\Theta} &= (\cdots((\hat{\boldsymbol{A}}_{\mathcal{K}}\hat{\boldsymbol{A}}_{\mathcal{K}})\hat{\boldsymbol{A}}_{\mathcal{K}})\cdots\hat{\boldsymbol{A}}_{\mathcal{K}})\hat{\boldsymbol{S}}^K \boldsymbol{H}^{(0)}\boldsymbol{\Theta} \\
&= \hat{\boldsymbol{A}}_{\mathcal{K}}\hat{\boldsymbol{S}}^K \boldsymbol{H}^{(0)}\boldsymbol{\Theta} \\
&= (\cdots((\hat{\boldsymbol{A}}_{\mathcal{K}}\hat{\boldsymbol{S}})\hat{\boldsymbol{S}})\cdots\hat{\boldsymbol{A}}_{\mathcal{K}})\boldsymbol{H}^{(0)}\boldsymbol{\Theta} \\
&= \boldsymbol{A}_{\mathcal{K}}\boldsymbol{H}^{(0)}\boldsymbol{\Theta}.
\end{aligned}$$

## C  Additional contents for experiments

### C.1  Downstream tasks

**Road function prediction** is a classification task that determines the functionality of a road. The label of the functionality is one of {"commercial", "construction", "education", "fairground", "industrial ", "residential", "retail", "institutional"} (`https://wiki.openstreetmap.org/wiki/Key: landuse`), which is derived from the neighborhood region (land use). The labels of road function are not from the road network data and were not considered in previous literature. In our experiments, we get the functionality from the land use data in OSM, while other data sources are also feasible. However, the functionality of regions is only available in several cities. Therefore, this task is very meaningful in generating labels and analyzing the urban status in a lot of cities.

**Road traffic inference** is a regression task predicting the average speed of vehicles on each road. It is widely used to evaluate the effectiveness of road representations in previous literature [2, 3, 23].

**Visual road retrieval** is a retrieval task, where the input is an image and underlying database stores road segments (e.g., a vector database of road embeddings). This task is to query roads where the input image should locate. A real-world scenario is that, a traveller want to visit some positions in a city. He / She gets some pictures, but may not know where they are located. This task can help them find those places in seconds.

### C.2  Baselines

- Deepwalk [29] is a network embedding algorithm to learn compressed vectorized node representations according to the structure of the graph. It first samples some random walks (i.e., sequences of nodes) from the graph. The nodes in random walks are regarded as words, while each random walk is regarded as a sentence. Then Deepwalk trains a skip-gram [24] model on the random walks and learns the representations of nodes. Deepwalk does not consider the features on each node.

- MVGRL [9] is a very powerful graph contrastive learning method. It generates another graph view via graph diffusion process. Then, it maximizes the mutual information between the original graph and the augmented graph, following a node-graph contrastive strategy. In our method, we incorporate the First Law based on this model.

- CCA-SSG [47] is an unsupervised graph representation method. Instead of using conventional mutual information estimators, CCA-SSG builds its contrastive loss upon Canonical Correlation Analysis. It does not need instance-level discrimination and negative sampling, and thus more efficient and scalable than contrastive based methods.

- GGD [51] is built upon the graph contrastive learning framework but replaces the traditional mutual information estimator with a "group discrimination" loss. The new loss does not require complicated mutual information estimation but only needs to discriminate whether a sample is from a positive or negative sample. Therefore, it is much faster than graph contrastive learning methods based on mutual information estimation.

- RFN [15] is a road network representation method based on graph attention networks [36]. It regards junctions (intersections of roads) as nodes to build a primal graph and its line graph, where road segments are nodes and connected road segments are linked. It then

designs relational fusion layers that can perform message passing between both graphs. However, this also significantly increases memory usage, especially on large graphs.

- SRN2Vec [39] is a the road network representation method based on random walk and skip-gram model [24]. To include geospatial information for road networks, it generates random walks according to both the graph topology and geospatial distance. To incorporate the basic features from OSM (e.g., road type), it introduces additional learning objectives such as classification for road type.

- SARN [2] is a road network representation framework based on graph contrastive learning. It builds a weighted adjacency matrix according to the road network topology, distance similarity, and angular similarity. It then follows GCA [57] to produce augmented graphs by dropping edges according to the weights. It also designs a negative sampling technique according to the distance. Finally, the model is trained on an InfoNCE loss [35].

## C.3 Settings and implementation details

We use the Adam optimizer [16] with the learning rate as 0.001 and set the training iterations as 2500 with early stopping. The sampling size is set as 4000. For the settings of baselines, we follow their default setting but set the dimension of representation as 512, the same as our method.

All the code is implemented with Python=3.11.8, PyTorch=2.1 (CUDA=11.8) [28], DGL=2.1 [41]. All the experiments are executed on a Ubuntu Server (Ubuntu 20.04), with $8 \times$ Nvidia Tesla V100 (32GB) GPUs, Intel(R) Xeon(R) Gold 6148 CPU @ 2.40GHz (40 cores and 80 threads) and 512 GB memory. The code of baselines is generally obtained from the authors' GitHub repo. The only exception is that we use the DGL's implementation (`https://github.com/dmlc/dgl/tree/master/examples/pytorch`) of Deepwalk and MVGRL for better efficiency.

## C.4 Scalability

In our experiments, we find that some of the previous works are not scalable in our datasets. The reason could be that previously, they used much smaller datasets. Specifically, the road networks in [3, 15, 39] have less than 10,000 nodes, less than one-tenth of our datasets, and thus, they do not need to consider the scalability issue. In our method, as we perform a sub-sampling process before applying graph convolution and loss, the memory usage on GPU does not grow with the data size, and thus, the proposed `Garner` is scalable to large road networks.

## C.5 Additional results on ablation studies

The ablation studies on Road Traffic Inference and Visual Road Retrieval are listed in Table 7 and Table 8 respectively.

Table 7: Ablation studies on Road Traffic Inference

| Methods | Singapore | | | NYC | | |
|---|---|---|---|---|---|---|
| | MAE ↓ | RMSE ↓ | MAPE ↓ | MAE ↓ | RMSE ↓ | MAPE ↓ |
| `Garner` - sns - aug - SVI | $3.04 \pm 0.04$ | $3.82 \pm 0.04$ | $0.629 \pm 0.041$ | $3.91 \pm 0.02$ | $5.16 \pm 0.03$ | $0.243 \pm 0.001$ |
| `Garner` - sns - aug | $2.99 \pm 0.02$ | $3.74 \pm 0.03$ | $0.610 \pm 0.036$ | $3.53 \pm 0.02$ | $4.70 \pm 0.04$ | $0.221 \pm 0.002$ |
| `Garner` - sns | $2.82 \pm 0.02$ | $3.54 \pm 0.03$ | $0.606 \pm 0.040$ | $3.35 \pm 0.02$ | $4.45 \pm 0.03$ | $0.210 \pm 0.002$ |
| `Garner` | $2.80 \pm 0.03$ | $3.52 \pm 0.04$ | $0.579 \pm 0.030$ | $3.30 \pm 0.02$ | $4.40 \pm 0.03$ | $0.207 \pm 0.002$ |

*"- sns" means without spectral negative sampling, but with feature shuffling to generate negative samples. "- aug" means without geographic configuration aware graph augmentation. "- SVI" means without street view images as inputs.*

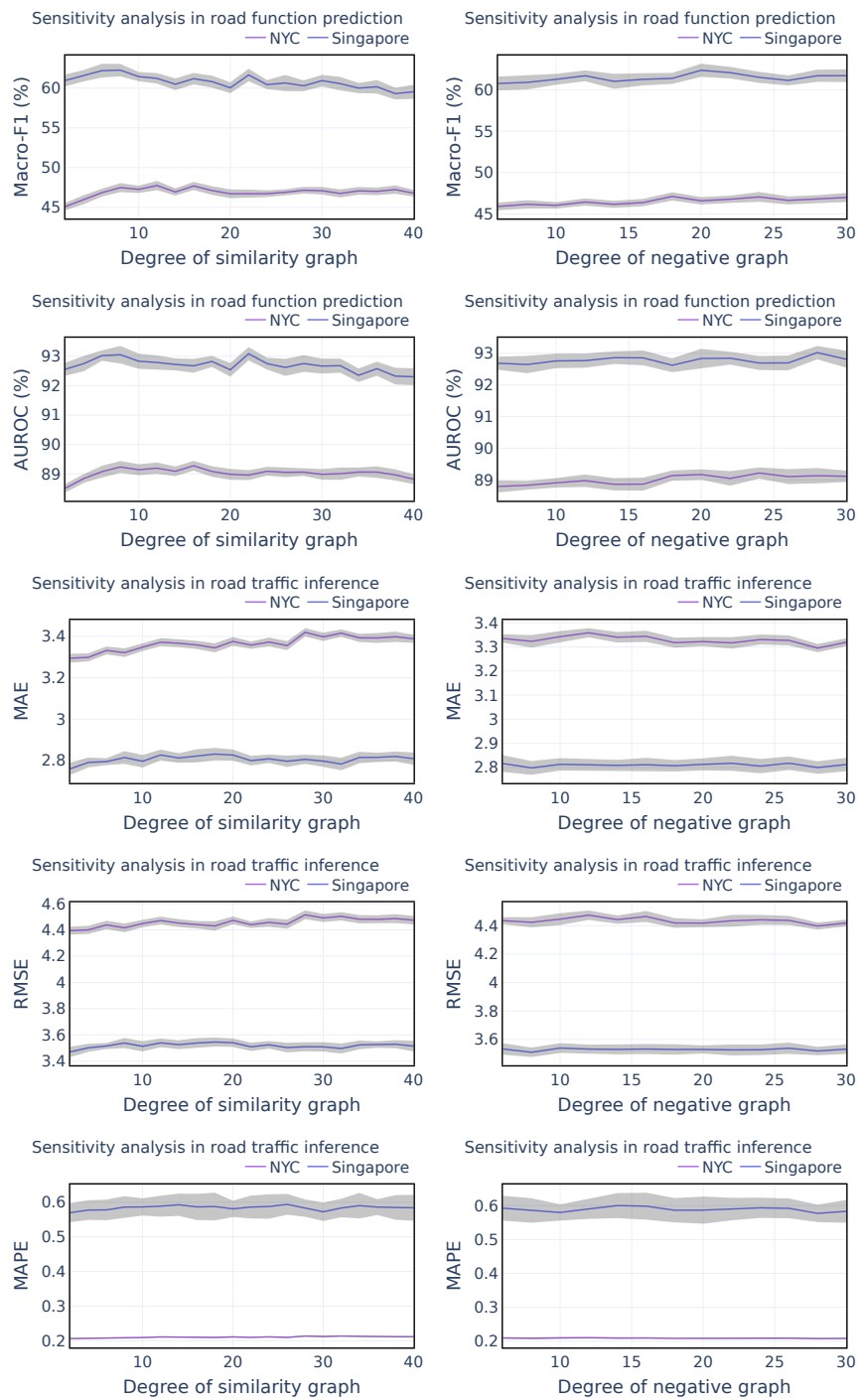

Figure 4: Sensitivity analysis of hyper-parameter $k$ and $d$.

Table 8: Ablation studies on Visual Road Retrieval

| Methods | Singapore | | NYC | |
|---|---|---|---|---|
| | Recall@10 ↑ | MRR ↑ | Recall@10 ↑ | MRR ↑ |
| `Garner` - sns - aug - SVI | 0.0088 | 0.0818 | 0.0324 | 0.1072 |
| `Garner` - sns - aug | 0.2002 | 0.3013 | 0.2803 | 0.2555 |
| `Garner` - sns | 0.3426 | 0.3112 | 0.4776 | 0.2805 |
| `Garner` | 0.4600 | 0.3387 | 0.5531 | 0.2985 |

*"- sns" means without spectral negative sampling, but with feature shuffling to generate negative samples. "- aug" means without geographic configuration aware graph augmentation. "- SVI" means without street view images as inputs.*

### C.6 Additional results on sensitivity analysis

## D   Further discussions

### D.1   Limitations

This paper is based on the Third Law of Geography and the Third Law of Geography. Though the two laws are generally true, the method in this paper may fail where the two laws are not applicable. For example, the First Law may fail on extremely large areas or limited data [55].

The potential negative societal impact includes: (1) Our method requires street view images (SVIs) along roads. However, SVIs may not be up-to-date and thus our methods may provide outdated information. Also, SVIs cannot provide everyday changes in a city. (2) our method currently does not consider adversarial attacks from data, and thus may provide incorrect information for downstream tasks if it is attacked.

### D.2   Broader Impact

As discussed in the introduction, road network representation learning provides fundamental instruments for various downstream tasks in urban computing. It can improve the traffic system in cities and enhance safety. It also provides essential references for urban planners who want to know various facets of the cities. We also admit that there could be some negative societal impacts. We are committed to ensuring our models are fair, unbiased, and respectful of individuals' privacy. We also acknowledge potential risks, such as misuse of the technology.

