# OpenReview forum: "Road Network Representation Learning with the Third Law of  Geography"
_NeurIPS.cc/2024/Conference — NeurIPS 2024 poster_

### Official Review · Reviewer_cFij · 2024-07-07

**Soundness:** 3
**Presentation:** 2
**Contribution:** 3
**Rating:** 6
**Confidence:** 4

**Summary:**

This paper proposes a novel framework for learning road network representations. Different from previous approaches, the proposed framework leverages a particularly designed graph contrastive learning method to integrate the Third Law of Geography into the process of road network representation learning, which can significantly alleviate the current shortcomings. In both classical and newly identified downstream tasks of road representation learning, the proposed framework achieves significant improvements compared with other state-of-the-art approaches.

**Strengths:**

S1. Novelty: The idea of integrating the Third Law of Geography in learning road network representations for urban computing tasks is novel.

S2. Technical solidity: The geographical law considered by the authors is mathematically modeled in the forms of graph augmentation and negative sampling, which is then integrated into the learning process of road network representation through a particularly designed graph contrastive learning framework. The proposal of this computational method lies in a non-trivial theoretical background.

S3. Comprehensiveness: The empirical validations of the proposed approach are comprehensive. Besides of verifying the proposed approach with classical tasks of urban computing, new learning tasks closely related to real scenarios are further identified by the authors and used in the experiments.

S4. Effectiveness: Compared with existing approaches, the proposed method achieves significant improvements on all downstream tasks.

**Weaknesses:**

W1. This paper lacks detailed explanations of using the Third Law of Geography in road network representation learning. The authors are suggested to give more explanations or examples to show the significance of using the Third Law.

W2. The spectral negative sampling is effective but may not be easy to understand for non-experts. The authors are encouraged to provide more explanations of this technique.

W3. In experiments, this paper only presents improvements in metrics, without more intuitive results. The authors are encouraged to give more discussions or empirical analysis of the implication of the third law on representations.

W4. The presentation can be improved. Also, there are some typos in the paper.

**Questions:**

As stated in “Weaknesses,” I have the following questions.

Q1. Are there more detailed explanations or examples that can show the significance of the Third Law of Geography in road network representation learning?

Q2. Are there more easy-to-understand descriptions/explanations of the proposed spectral negative sampling method?

Q3. Are there case studies or discussions in the experiments that can show the implication of the Third Law of Geography in road network representation learning?

Q4. Since this work is based on the Third Law of Geography, it seems it is similar to context-aware spatiotemporal learning. It is encouraged for authors to discuss the correlations and distinguishment between context-based ST learning and Third Law of Geography based learning.

**Limitations:**

The authors have discussed the potential limitations but not much sufficient.

---

> ### Author Rebuttal · Authors · 2024-08-07
>
> Thank you for your constructive comments and suggestions. We apologize for any confusion. We are pleased to inform you that all concerns have been addressed. Below are our responses to each comment.
>
> Responses to Weaknesses:
>
> 1. To illustrate the Third Law of Geography, consider two roads surrounded by residential buildings. Even though two roads are located very far apart, according to the third law, as they have very similar geographic configurations, they should still have similar representations. We will include these discussions in the final version of the paper to provide readers with easy understandings of the proposed method.
> 2. In the "Spectral Negative Sampling" section, we design a negative sample compared with the positive sample (the augmented graph from the third law of geography) for better contrastive learning. The positive sample connects nodes with very similar geographic configurations. Thus if we design a negative sample which mainly connects roads with dissimilar geographic configurations, then by discriminating the anchors (the original graph) from the negative sample, we can enhance the representations and achieve that "roads with dissimilar geographic configurations should have dissimilar representations". Then we design the specific form of the negative sample, which is inspired by the sparsest cut problem (Equation 8). In particular, it could be a complete graph with a proper graph encoder (as in Section 4.5). However, leveraging the complete graph as the negative sample is very time and space consuming (, even with subgraph sampling). Then we apply spectral graph sparsification to effectively approximate the complete graph, which results in a d-regular graph. Finally, we design a d-regular graph as the negative sample, with a properly designed graph encoder.
>
> 3. Following the reviewer's suggestion, we conducted a case study comparing the First and Third Laws of Geography. The case study is included in the attached PDF under "Author Rebuttal by Authors" and will be added to the final version of the paper.
>     - We randomly selected an anchor road, computed its representation similarity with other roads (according to cosine similarity), and displayed the top 10 most similar roads. The anchor road is red, the top 10 similar roads by the First Law are blue, and those by both laws are orange.
>     - With only the First Law, similar roads are much closer to the anchor. With both laws, similar roads are farther but have similar geographic configurations, as shown by comparing street view images. This demonstrates that the Third Law ensures similar representations for roads with similar configurations, regardless of distance.
>
> 4. Thanks for your suggestion. We will polish and double-check the writing.
>
>
>
> Response to the Questions:
>
> 1. (Refer to response to weakness 1)
> 1. (Refer to response to weakness 2)
> 1. (Refer to response to weakness 3)
> 4. Following the suggestion of the reviewer, we discuss the correlations and differences between: (1) context and geographic configuration; and (2) context-aware spatiotemporal learning and the Third Law of Geography.
>    - (1) **Context** is usually defined as the some contents or attributes of a spatial entity, such as the spatial context of a road segment in [1] and the context of a POI [3], where the spatial context is still some attributes of a road. "**Geographic configuration**" is defined as "the makeup and the structure of geographic variables over some spatial **neighborhood** around a point," in A-Xing Zhu's paper [2] on the Third Law of Geography. In our problem, the geographic configuration of a road segment includes both its features and the features of its **neighborhood**, e.g., its surrounding buildings, natural environments, regions, etc. And street view images are very good proxies to describe those. In a word, the major difference is that the term context, in the literature of spatiotemporal learning, focuses only on a spatial entity itself, while **geographic configuration** concerns both the attributes of the spatial entity itself and its spatial neighborhood.
>    - (2) **Context aware learning**, inspired by word2vec [4], encourages entities within a context to have similar representations, which is more similar to the First Law of Geography, "everything is related to everything else, but near things are more related than distant things". In contrast, the **Third Law of Geography** states that "The more similar geographic configurations of two points (areas), the more similar the values (processes) of the target variable at these two points (areas)." It maps the similarity relationship of the geographic configurations to the similarity relationship of the target variable (i.e., road representation in our paper).
>
>    We will include the above discussions in the final version of the paper.
>
>
> ---
>
> [1] Robust Road Network Representation Learning: When Traffic Patterns Meet Traveling Semantics. In CIKM '21.
>
> [2] Spatial prediction based on Third Law of Geography. Annals of GIS, 2018.
>
> [3] How is the Third Law of Geography different? Annals of GIS, 2022.
>
> [4] Distributed representations of words and phrases and their compositionality. In ICLR.

---

> ### Author Response · Authors · 2024-08-10
> **Additional Questions**
>
> Dear Reviewer cFij,
>
> Thanks very much for providing the constructive and motivating feedback! Can you please let us know whether we have addressed all your questions and whether you have any additional feedback?
>
> Thank you!

---

> > ### Comment · Reviewer_cFij · 2024-08-14
> > **Thanks for rebuttal**
> >
> > I would like to thank the authors for their detailed responses, and my concerns are well addressed.

---

### Official Review · Reviewer_u7JA · 2024-07-11

**Soundness:** 2
**Presentation:** 2
**Contribution:** 2
**Rating:** 3
**Confidence:** 4

**Summary:**

This paper introduces a novel framework for road network representation learning, with the key innovation being the incorporation of the Third Law of Geography. This concept is implemented through a tailored graph contrastive learning objective, featuring geographic configuration-aware graph augmentation and spectral negative sampling. To further preserve geographic proximity (the First Law of Geography), which has been proven crucial in existing literature, the authors further propose a dual contrastive learning strategy to accommodate both modeling perspectives. Experimental results on datasets from Singapore and New York City demonstrate significant improvements over several baseline methods in road function prediction, road traffic inference, and visual road retrieval tasks.

**Strengths:**

1. The paper addresses a significant problem in road network representation learning, which has important implications for real-world applications such as traffic management and urban planning.

2. The core idea of the paper is grounded in established geographical theory, and the incorporation of street view images as a supplementary data source enhances the implementation of this concept.

3. The authors demonstrate awareness of scalability issues and incorporate sampling techniques in their method design, enabling efficacy in large-scale road networks.

4. The paper presents comprehensive experimental validations of the model's superiority over three downstream tasks.

**Weaknesses:**

1. The research challenges identified by the paper lack depth. In particular, the first challenge is insufficiently articulated, equivalent to stating that "generating road representations that preserve the Third Law of Geography is challenging".

2. A significant weakness of the paper lies in the limited novelty of its ideas and methodological design:

   a. While existing literature may not explicitly introduce the concept of the Third Law of Geography, several works have already considered the similarity of road segments in regions with similar functionalities [1,2,3]. Moreover, in the broader field of spatio-temporal data mining, several studies have considered the hierarchical structure of urban road networks [4,5].

   b. The simple concatenation of street view image representations with other road features lacks sophistication. Images and other road attributes exist in different modalities, potentially introducing multi-modal data fusion issues that may impede the effective encoding of street view images into road representations.

   c. The Simple Graph Convolution (SGC), its theoretical interpretation, and graph contrastive learning have already been well-developed in previous research.

   d. The fusion of the third law and the first law is presented as one of the two major challenges the paper aims to address. However, the proposed solution merely combines two contrastive learning tasks without a thorough discussion of potential conflicts between them.

   e. Although the paper considers scalability issues, the proposed sub-sampling solution is relatively straightforward.

3. The paper lacks detailed discussions on the universality of the third law and the first law, particularly in large-scale road networks, raising concerns about the practical applicability of the method. Besides, merely mentioning this in the limitations section is insufficient.

4. The second paragraph at the beginning of section 4 contains confusing notations that are not previously introduced.

5. Experimental design:

   a. As the paper's solution to improve scalability is straightforward, the baseline methods should also be equipped with the same graph sampling techniques to ensure a fair comparison.

   b. The paper lacks case studies that demonstrate the effectiveness of integrating street view images. Relying solely on ablation studies is inadequate for this purpose.

   c. The paper does not use weighted average of the two contrastive learning tasks, which is important as the first law and the third law might conflict in certain scenarios. Along this line, experiments on the trade-off between the two contrastive learning tasks should also be conducted.

[1] Pre-training Context and Time Aware Location Embeddings from Spatial-Temporal Trajectories for User Next Location Prediction. In AAAI 2021.

[2] Pre-training local and non-local geographical influences with contrastive learning. Knowledge-Based Systems 2023.

[3] Pre-training Contextual Location Embeddings in Personal Trajectories via Efficient Hierarchical Location Representations. In ECML-PKDD 2023.

[4] Semi-Supervised Hierarchical Recurrent Graph Neural Network for City-Wide Parking Availability Prediction. In AAAI 2020.

[5] GPT-ST- Generative Pre-Training of Spatio-Temporal  Graph Neural Networks. In NIPS 2023.

**Questions:**

1. Why is the spectral negative sampling strategy only applied to the contrastive learning task corresponding to the third law? How does the derivation of this strategy in section 4.5 relate to the mutual information maximization in equation (5)? Would it not be more intuitive to design an additional contrastive learning task to focus on encouraging the node representations of the augmented graph to satisfy both the third law and its inverse version?

2. Why does the inverse version of the third law in section 4.5 hold? For instance, two roads with dissimilar geographic configurations might still exhibit spatial proximity, suggesting that their representations should remain similar.

3. What proportion of roads in the dataset are associated with street view images? How does the model perform in geographic regions where (a) street view images are limited or of poor quality, or (b) the two geographic laws do not hold?

4. Compared to widely used Points of Interest (POI) data, what are the unique advantages of street view images in this context?

5. What are the specific experimental settings for the road traffic prediction task? If the objective is to predict future traffic variations, how do static road network representations contribute to this task without considering the dynamic correlations among road segments?

**Limitations:**

see weakness

---

> ### Author Rebuttal · Authors · 2024-08-07
>
> Thank you for your detailed comments and suggestions. We apologize for any confusion. We are pleased to inform you that all concerns have been addressed. Below are our responses to each comment. As the reviewer has many concerns, we conduct more discussions to address them.
>
> Response to Weaknesses:
>
> 1. We analyze the meanings of the third Law and point out that incorporating it into the road network representation learning is very beneficial. This challenge is in contrast with the first Law, which emphasizes the importance of distance effect. The Third Law focuses on mapping between relationships for geographical configurations in geospatial entities (road segments) to relationships of target variables (representations). It is more challenging to learn the mapping of relationships than to learn the mapping of values. More explanations can be found in our response to "Weakness 2" below.
>
> 2. We argue the **novelty** of our idea and method as follows. In general, the novelty of our papers is twofold: (1) we introduce the Third Law of Geography for road network representation learning; (2) we design novel methods to learn the third law in graph neural networks (GNNs). In particular, our **novel designs** include geographic configuration-aware graph augmentation and spectral negative sampling. The responses to the detailed questions are listed as follows.
>    - a. The term "geographic configuration" is not functionalities of regions, and the Third Law of Geography is not related to hierarchical structures in cities. In particular, the definition of "geographic configuration" in [1] is "the makeup and the structure of geographic variables over some spatial **neighborhood** around a point.", far beyond the functionalities of regions. The street view images (SVIs), which provide the visual features of a road, also include important descriptions of a road. For example, the surface condition of a road and the number of lanes in a road (, which is not available for the majority of OSM data,). Also, the geographic configuration includes descriptions of a point itself and its neighborhood. Street view images describe both the visual features of a road and its **neighborhood**, e.g., its surrounding buildings, regions etc. Thus we think the street view images are a good proxy, among various current data, to describe the geographic configuration of a road. Besides, our work is **not** related to hierarchical structures in a city. However, we would also like to add another paragraph in the "Related Work" section to discuss the literature that the reviewer mentions.
>
>    - b. For multi-view / modality fusion, our design is not a direct concatenation of different data. Our designs include: (1) projecting different data into another space with learnable parameters and then concatenating them (as illustrated in Section 4.1); (2) fusing information from multiple views with mutual information maximization. The first design (joint representation) is widely adopted in multi-modality fusion [8], while the second design is inspired by [9]. Both [8] and [9] demonstrate that information from different views can be fused properly in our method.
>
>    - c. Our main contributions, as listed at the end of the Introduction in the paper, are: (1) introduction of the **Third Law of Geography** to road network representation; (2) **geographic configuration-aware graph augmentation** and **spectral negative sampling** in the graph contrastive learning framework. Those are our **original** designs and have never appeared in previous literature. As our proposed method is based on graph contrastive learning and SGC, which are preliminaries of our method, we need to put the contents in the method for those who are not familiar with these topics. Besides, the theoretical interpretation of SGC is closely related to the implementation of the Third Law of Geography, and thus we discuss why SGC can achieve our goal in Section 4.3. We believe that bridging existing theories to real-world applications is also very important and can extend the scope of existing theories.
>
>    - d. We have two efforts on fusing the First and the Third Law (the tuning of weights are listed in the comments):
>
>        - The two losses have some sharing parameters (module $g_{\theta_0}$), which can learn the consensus, while those modules ($g_{\theta_1}$ and $g_{\theta_2}$) that do not have sharing parameters learn the discrepancy (potential conflicts). The final representation is the aggregation of the outputs from the three modules.
>        - Train the two losses jointly. For joint training, in our preliminary experiments, we have tried some multi-objective optimization techniques to balance the two terms. Specifically, we have tried the Pareto optimum [1] in neural networks to adaptively choose proper weights to balance the two terms. However, we find that (1) with this technique, the results do not have significant improvement; (2) We find the optimal weights for the two terms vary from (0.4, 0.6) to (0.6, 0.4), very close to (0.5, 0.5). Those are the reasons why we do not tune the weights and state in the last paragraph of Section 4.6 that "Therefore, we do not introduce additional hyper-parameters to adjust their weights."
>        - We also conduct some empirical results on tuning the weights of the two losses, and the results on the road function prediction task are listed as follows. We introduce another hyper-parameter $\beta$ to balance the loss of the third law ($\mathcal{L}_1$) and the first law ($\mathcal{L}_2$) : $\mathcal{L} = \beta \mathcal{L}_1 + (1 - \beta) \mathcal{L}_2$ . The results show that the learning performance is not sensitive to the weight $\beta$.
>    - e. Our paper does not focus on scalability issues in road network representation learning, and we do not claim our contribution on the scalability issue neither. We provide a sub-sampling trick in our implementation, which works well within our framework.

---

> ### Author Response · Authors · 2024-08-07
> **[Rebuttal by Authors - 2] Results of Tuning the Weights of Losses**
>
> **Road Function Prediction on Singapore**
>
> | $\beta$ | Micro-F1 (%) $\uparrow$ | Macro-F1 (%) $\uparrow$ | AUROC (%) $\uparrow$ |
> | ------- | ----------------------- | ----------------------- | -------------------- |
> | 0.1     | 81.23 $\pm$ 0.36        | 62.03 $\pm$ 0.84        | 92.90 $\pm$ 0.28     |
> | 0.2     | 81.48 $\pm$ 0.37        | 62.61 $\pm$ 0.85        | 93.01 $\pm$ 0.24     |
> | 0.3     | 81.22 $\pm$ 0.34        | 62.11 $\pm$ 0.62        | 92.91 $\pm$ 0.25     |
> | 0.4     | 80.99 $\pm$ 0.30        | 61.50 $\pm$ 0.70        | 92.66 $\pm$ 0.26     |
> | 0.5     | 81.40 $\pm$ 0.30        | 62.45 $\pm$ 0.64        | 93.27 $\pm$ 0.22     |
> | 0.6     | 81.02 $\pm$ 0.44        | 60.94 $\pm$ 0.85        | 92.69 $\pm$ 0.21     |
> | 0.7     | 81.19 $\pm$ 0.37        | 62.26 $\pm$ 0.88        | 92.82 $\pm$ 0.21     |
> | 0.8     | 81.26 $\pm$ 0.33        | 61.91 $\pm$ 0.83        | 92.81 $\pm$ 0.23     |
> | 0.9     | 81.40 $\pm$ 0.40        | 62.54 $\pm$ 0.72        | 93.05 $\pm$ 0.24     |
>
> **Road Function Prediction on NYC**
>
> | $\beta$ | Micro-F1 (%) $\uparrow$ | Macro-F1 (%) $\uparrow$ | AUROC (%) $\uparrow$ |
> | ------- | ----------------------- | ----------------------- | -------------------- |
> | 0.1     | 82.83 $\pm$ 0.21        | 46.70 $\pm$ 0.45        | 89.18 $\pm$ 0.21     |
> | 0.2     | 82.93 $\pm$ 0.19        | 47.17 $\pm$ 0.37        | 89.24 $\pm$ 0.17     |
> | 0.3     | 82.91 $\pm$ 0.20        | 47.25 $\pm$ 0.57        | 89.17 $\pm$ 0.20     |
> | 0.4     | 82.86 $\pm$ 0.20        | 47.08 $\pm$ 0.49        | 89.14 $\pm$ 0.19     |
> | 0.5     | 82.97 $\pm$ 0.16        | 47.22 $\pm$ 0.42        | 89.30 $\pm$ 0.21     |
> | 0.6     | 82.97 $\pm$ 0.23        | 46.80 $\pm$ 0.56        | 89.22 $\pm$ 0.15     |
> | 0.7     | 83.04 $\pm$ 0.15        | 47.57 $\pm$ 0.52        | 89.21 $\pm$ 0.20     |
> | 0.8     | 82.89 $\pm$ 0.21        | 46.76 $\pm$ 0.47        | 89.03 $\pm$ 0.15     |
> | 0.9     | 82.85 $\pm$ 0.23        | 46.49 $\pm$ 0.47        | 88.97 $\pm$ 0.20     |

---

> ### Author Response · Authors · 2024-08-07
> **[Rebuttal by Authors - 3] Response to Weaknesses - Continued**
>
> 3. The effectiveness and when the third law and the first law are expected to work have been extensively discussed in previous literature [10, 11] in geographical sciences. Our paper applies the Third Law of Geography to road network representation rather than examining the Third Law itself. We would like to recommend readers to refer [11] for more details.
>
> 4. Sorry for this confusion. The meaning of those three notations is explained here. $\boldsymbol{Z}$ is the output of some GNN. $\boldsymbol{Z}^{[0]}$, $\boldsymbol{Z}^{[1]}$ and $\boldsymbol{Z}^{[2]}$ are three outputs from different GNNs as illustrated in Fig. 1, where the number $i$ in the superscript $\cdot^{[i]}$ indicates that $\boldsymbol{Z}^{[i]}$ is the output of $i$-th GNN.
>
> 5. About the experimental design:
>
>    - a. Thanks for your suggestion. First, most baselines have their own strategies for tackling the scalability issues. These strategies are proposed according to the peculiarities of those methods. Thus it is **not realistic** to equip each baseline with the sub-sampling technique. Second, other baselines, such as SRN2Vec, have unrelated model structures regarding sampling, which prevents the sub-sampling trick from being applied in their methods. Third, to explore the best performances of other approaches, we did not integrate our method with other baselines and use their recommended settings in the experiments.
>    - b. Following the reviewer's suggestion, we conducted a case study comparing the First and Third Laws of Geography. The case study is included in the attached PDF under "Author Rebuttal by Authors" and will be added to the final version of the paper. (1) We randomly selected an anchor road, computed its representation similarity with other roads (according to cosine similarity), and displayed the top 10 most similar roads. The anchor road is red, the top 10 similar roads by the First Law are blue, and those by both laws are orange. (2) With only the First Law, similar roads are much closer to the anchor. With both laws, similar roads are farther but have similar geographic configurations, as shown by comparing street view images. This demonstrates that the Third Law ensures similar representations for roads with similar configurations, regardless of distance.
>    - c. We have discussed this issue in "2.d."

---

> ### Author Response · Authors · 2024-08-07
> **[Rebuttal by Authors - 4] Response to Questions**
>
> 1. Sorry for the confusion. We indeed use one contrastive learning objective to incorporate the Third Law of Geography. Contrastive learning, with its mathematical support from mutual information maximization [2], requires both positive samples and **negative samples** to learn its loss function (Eq. (4) (5)) [3, 4]. Section 4.5 demonstrates how to generate a negative sample for graph contrastive learning. Positive samples provide information that is positive to the anchor (the original graph), while negative samples provide negative information for contrast. By contrasting with both positive samples and negative samples, models can learn good representations without supervision [5]. Following this principle, we design a positive sample where roads with very similar geographic configurations are connected, and also design a negative sample, in Section 4.5 Spectral Negative Sampling, where roads with very dissimilar geographic configurations are connected. The detailed design of the negative sample is inspired by the sparsest cut problem (in spectral graph theory) and further optimized by spectral graph sparsification. This is why the subtitle of this subsection is "Spectral Negative Sampling". As the spectral negative sampling in this paper is particularly designed to learn the third law, we only prepare it for the third law. According to the principle of contrastive learning, it is also intuitive to incorporate the inversion version of the third law by negative samples.
> 2. If the first law (spatial proximity) and the third law (geographic configuration) give different results on similarity, the proposed model can make some tradeoffs between the two laws, when learning road representations. Possible scenarios include spatially distant roads with similar geographic configurations and spatially proximal roads with different geographic configurations. And this is why we need to "jointly consider the third law of geography and the first law of geography" mentioned by Reviewer UWD1.
> 3. We sample street view images (SVIs) on each road. In practice, more than 95% of roads are associated with SVIs. For those roads without SVIs associated with them, we leverage SVIs within several meters (i.e., SVIs located in the buffer zone of a road) as proxies to represent their geographic configurations. Finally, every road has several SVIs associated on with it. Although the scenarios mentioned by the reviewer are not the critical issues tackled by the current work, if the street view data are limited or the dataset contains some samples that do not follow the two geographic laws, the proposed method might face challenges in performing robustly as other related approaches do. However, these issues can be potentially addressed by designing the new version of Garner to endow it with the capability of learning with incomplete or contaminated road network data. In the future, we will keep on exploring the effectiveness of the proposed Garner by considering the valuable suggestions raised by the reviewer. We will also include the above discussions in the final version of the paper to provide readers who are interested in the proposed Garner with more possibilities to improve it.
> 4. The unique advantages of street view images (SVI) are listed as follows:
>    - **Distribution and Coverage of POIs**:  A major issue of POIs in our problem is that they are unevenly distributed in cities, and a large portion of road segments do not have nearby POIs. In contrast, SVIs cover almost every road in a city, and we can sample SVIs for each road segment.
>    - **Commercial Bias of POIs**: Another issue is that POIs are provided by Internet Web Maps for map search, and thus their contents are mainly for commercial functions (e.g., restaurants), leading to poor performance in non-commercial regions such as industrial or residential areas [12, 13]. However, there are many roads outside commercial regions, where POIs lack a lot of information. In contrast, SVIs can provide meaningful information across diverse regions.
> 5. Following previous work [6, 7], the "Road Traffic Inference", a standard downstream task to evaluate the effectiveness of road representations, is to predict the average speed of each road, not to predict the future traffic. For static representation learning, all of the existing studies on road network do not consider the setting of predicting future traffic variations. Developing an advanced model which can effectively capture the dynamic correlations based on this paper would be our future work.

---

> ### Author Response · Authors · 2024-08-07
> **[Rebuttal by Authors - 5] References of Author Rebuttal and Response to Reviewer Comments**
>
> [1] Multi-Task Learning as Multi-Objective Optimization. In NeurIPS 2018.
>
> [2] MINE: Mutual Information Neural Estimation. In ICML 2018.
>
> [3] Representation Learning with Contrastive Predictive Coding.
>
> [4] Deep Graph Infomax. In ICLR 2019.
>
> [5] Understanding Contrastive Representation Learning through Alignment and Uniformity on the Hypersphere. ICML 2020.
>
> [6] Robust Road Network Representation Learning: When Traffic Patterns Meet Traveling Semantics. CIKM 2021.
>
> [7] Relational Fusion Networks: Graph Convolutional Networks for Road Networks. IEEE Trans. Intell. Transport. Syst. 2021.
>
> [8] Multimodal Machine Learning: A Survey and Taxonomy. IEEE Transactions on Pattern Analysis and Machine Intelligence, 2019.
>
> [9] Contrastive multi-view representation learning on graphs. In ICML 2020.
>
> [10] Spatial prediction based on Third Law of Geography. Annals of GIS, 2018.
>
> [11] How is the Third Law of Geography different? Annals of GIS, 2022.
>
> [12] Urban2vec: Incorporating street view imagery and pois for multi-modal urban neighborhood embedding. In AAAI 2020
>
> [13] Knowledge-infused Contrastive Learning for Urban Imagery-based Socioeconomic Prediction. In WWW 2023

---

> ### Author Response · Authors · 2024-08-10
> **Additional Questions**
>
> Dear Reviewer u7JA,
>
> Thanks very much for providing the constructive and motivating feedback! Can you please let us know whether we have addressed all your questions and whether you have any additional feedback?
>
> Thank you!

---

> > ### Comment · Reviewer_u7JA · 2024-08-11
> > **Acknowledgement**
> >
> > I thank the authors for their detailed explanation. However, the responses on the novelty of the method and the argument on scalability are not very convincing. I would like to keep my score.

---

> ### Author Response · Authors · 2024-08-14
> **Further Clarification of Novelty - 1**
>
> Thanks for your reply. Could you please kindly let us know whether we have addressed some of your concerns?
>
> We would like to do some further clarification on the novelty. In summary, this work is the first attempt to consider the **Third Law of Geography** for the **road network representation learning task**.
> 1. The meaning of geographic configuration in the **Third Law of Geography** is far beyond functionalities of regions. The definition of "geographic configuration" in [6] is "the makeup and the structure of geographic variables over some spatial neighborhood around a point". The makeup and structure of a road contains a lot of information such as its surrounding buildings, its width, its surrounding region. The richness of this information cannot be described with a simple "functionality of region." For example, the height of the surrounding building of a road will influence the traffic on a road [9], but the height is not some functionality of a region.
> 2. Existing literature [1, 2, 3, 4, 5] does not mention the **Third Law of Geography** and the incorporation of the third law into neural networks. The literature listed by the reviewer still only considers the **First Law of Geography**. As illustrated in the Introduction of our paper, current literature make the geospatial entities within a context (spatial neighborhoods) to be similar. In contrast, the third law does not consider any spatial distance, and it argues that spatial entities, with similar geographic configurations, should be similar, even if they are very far apart.
>     - [1] considers spatial context on condition of specific functionality. It still focuses on (conditional) spatial proximity.
>     - [2] considers "ncorporate geospatial proximity as a local geographical influence and relative distance differences as a non-local geographical influence" [2], which still depends on spatial proximity and distance.
>     - [3] considers context, which includes a grid at a larger scale (spatial regions) and sequences (temporal sequences). But it still considers spatial proximity.
>     - [4, 5] consider hierarchical structures in cities, which are not related to our work.
>     - Spatial proximity and distance, as demonstrated in the Introduction of our paper, is depicted by the **First Law of Geography**. In contrast, the **Third Law of Geography** does NOT consider any spatial proximity (including distance) between two target entities, it sololy depends on geographic configurations.
> 3. In our method design to incorporate the third law: **geographic configuration-aware graph augmentation** and **spectral negative sampling**, they do NOT consider any spatial proximity or spatial distance between two road segments. This is why our method is able to generate similar representations for roads with very similar geographic configurations, even though they are very far apart. A case study to illustrate this has been included in the attached PDF under "Author Rebuttal by Authors".
> 4. The papers that you list are not doing the same **research problem** as ours. The research problem in our paper, as stated in the title is **road network representation learning**. In particular,
>     - [1] is to solve "Next Location Prediction".
>     - [2] is to solve "next POI recommendation" task.
>     - [3] is to learn "location embedding" to solve "Next Location Prediction" and "trajectory classification". "Locations", defined in [3] are grids (or regions) in cities. Grids are pologons. In contrastive, a road a polyline, which does not surround some area.
>     - [4] is to solve "Parking Availability Prediction".
>     - [5] is to solve traffic prediction on sensor network [7], not road network.
>     - Our paper is to solve "road network representation learning", and the downstream tasks include: road function prediction, road traffic inference and visual road retrieval. Both our pre-training task and downstream tasks are totally different from the papers you list.
>     - The word "road" does not appear in the main body of [1, 3, 5]. The word "road" nearly does not appear in [2, 4].
>
>
> ---
>
> [1] Pre-training Context and Time Aware Location Embeddings from Spatial-Temporal Trajectories for User Next Location Prediction. In AAAI 2021.
>
> [2] Pre-training local and non-local geographical influences with contrastive learning. Knowledge-Based Systems 2023.
>
> [3] Pre-training Contextual Location Embeddings in Personal Trajectories via Efficient Hierarchical Location Representations. In ECML-PKDD 2023.
>
> [4] Semi-Supervised Hierarchical Recurrent Graph Neural Network for City-Wide Parking Availability Prediction. In AAAI 2020.
>
> [5] GPT-ST- Generative Pre-Training of Spatio-Temporal Graph Neural Networks. In NIPS 2023.
>
> [6] Spatial prediction based on Third Law of Geography. Annals of GIS, 2018.
>
> [7] Diffusion Convolutional Recurrent Neural Network: Data-Driven Traffic Forecasting. In ICLR.
>
> [9] Street view imagery in urban analytics and gis: A review. Landscape and Urban Planning.

---

> > ### Author Response · Authors · 2024-08-14
> > **Further Clarification of Novelty - 2**
> >
> > For the scalability and efficiency issue, we have two designs: subgraph sampling and spectral graph sparsification. The spectral graph sparsification for our spectral negative sampling is our special design and cannot be applied to other methods.
> > 1. The subgraph sampling is to sample a much small graph from the huge road network.
> > 2. Even with subgraph sampling, the negative sampling to encourage "the reverse implication of the Third Law: roads with dissimilar configurations should have dissimilar representations" is still very time and space consuming. It requires to sample a complete graph, with time and space complexity of $O(|V^{\prime}|^2)$, where $|V^{\prime}|$ is the number of nodes in the sampled subgraph. To improve the efficiency and scalability, we follow the theoretical results of spectral graph sparsification [8], to approximate the complete graph with a $k$-regular graph, which has much less edges, with complexity of $O(|k V^{\prime}|)$, where $d$ is the degree of nodes, and we set $k=6$ in our experiments. It is non-trival to a adopt theoretical result to negative sampling.
> > 3. The results of spectral sparsification in our paper, is only application to approximate a complete graph. Thus it cannot be applied to other baselines, which either do not have negative sampling or do not use complete graph in negative sampling.
> >
> >
> > Thank you again to providing constructive feedbacks? Can you please let us know whether we have addressed all your questions and whether you have any additional feedback?
> >
> >
> > ---
> > [8] Daniel A. Spielman and Shang-Hua Teng. Spectral sparsification of graphs. SIAM J. Comput.

---

### Official Review · Reviewer_ggbs · 2024-07-12

**Soundness:** 3
**Presentation:** 3
**Contribution:** 3
**Rating:** 5
**Confidence:** 3

**Summary:**

This paper proposes a novel method for learning representations of road networks. It highlights the limitations of existing methods that primarily use the First Law of Geography, which emphasizes spatial proximity. The authors introduce a new framework, Garner, that incorporates the Third Law of Geography, focusing on geographic configuration similarities. They employ a graph contrastive learning approach with geographic configuration-aware graph augmentation and spectral negative sampling. The framework is evaluated on real-world datasets from Singapore and New York City, showing significant performance improvements in downstream tasks.

**Strengths:**

1.	The paper introduces a novel application of the Third Law of Geography, providing a fresh perspective in the field of geographic information systems. The methodology is robust, with thorough theoretical grounding and comprehensive empirical evaluation.
2.	The paper provides a solid theoretical basis for its methods. The mathematical proofs are thorough and well-constructed
3.	The proposed approach demonstrates substantial improvements in real-world datasets, highlighting its practical value.

**Weaknesses:**

1.	In Section 4.2, the kNN and threshold-based methods for building new connections are highly sensitive to their parameters. This sensitivity greatly impacts the generation of new connections, making it difficult to control and optimize, thereby affecting the model's consistency and robustness.
2.	The study is limited to datasets from Singapore and New York City; additional testing on more diverse datasets would enhance the findings' applicability.
3.	There is no code or dataset provided.

**Questions:**

1.	How does the proposed method perform on road networks with significantly different characteristics (e.g., rural vs. urban)? As the authors said in section 4.5: "roads with dissimilar geographic configurations should have dissimilar representations."
2.	Are there any examples or case studies demonstrating the practical implementation of Garner in different urban contexts?
3.	What are the computational requirements and scalability aspects of the proposed method for larger datasets? For example, if we want to measure the similarity between different cities, we need to use national or global maps.
4.	How does the model deal with dynamically changing road networks, such as those undergoing construction or frequent changes? Does the whole model need to be re-trained after the changes?

**Limitations:**

1.	The authors should consider testing their method on a more diverse set of datasets to ensure broader applicability.
2.	Consideration of temporal changes in road networks and how the model adapts to these changes would strengthen the practical application of the method.

---

> ### Author Rebuttal · Authors · 2024-08-07
>
> Thank you for your constructive comments and suggestions. We apologize for any confusion. We are pleased to inform you that all concerns have been addressed. Below are our responses to each comment.
>
> Response to the Weaknesses:
>
> 1. We have conducted the sensitivity test on the hyper-paramter $k$ of kNN graph, and we find our method is not sensitive to $k$.
>     - For implementation, we choose the kNN graph because of its high efficiency.
>     - For kNN graphs, the hyper-parameter is $k$. Therefore, we conducted comprehensive hyper-parameter sensitivity tests on various datasets and metrics, and the results are reported in Section 5.3 and Appendix C.6. The results show that the $k$ does **not** have a significant impact the performance of various downstream tasks. In other words, our model is not sensitive to the values of $k$. In our main experiments, we also use $k=6$ on all experiments, and this setting can produce much better results than all the baselines.
>
> 2. Thanks so much for your suggestion. Currently, we only use those datasets because of the limited data sources. Specifically, street view images are very **expensive** and time-consuming to collect, and other sources of data (e.g., road function, and traffic speed) are unavailable in most cities. To mitigate the potential issues caused by data availability, we have considered diverse downstream real-world tasks, i.e., road function prediction, road traffic inference, and visual road retrieval, to comprehensively test the proposed method and other baselines. In the future, we will try to collect more datasets for evaluation.
>
> 3. Sorry for the confusion, we have uploaded the code in the "Supplementary Material" under "Abstract" in this webpage.
>
>
>
> Response to Questions:
>
> 1. We achieve "roads with dissimilar geographic configurations should have dissimilar representations." by negative sampling in the contrastive learning framework. In particular, contrastive learning requires positive samples and negative samples. Thus we design a negative graph $\bar{\mathcal{G}}^{[1]}$ whose edges mainly connect roads with dissimilar geographic configurations. By discriminating the negative sample and anchor, we can achieve "roads with dissimilar geographic configurations should have dissimilar representations."
>
> 2. Following the reviewer's suggestion, we conducted a case study comparing the First and Third Laws of Geography. The case study is included in the attached PDF under "Author Rebuttal by Authors" and will be added to the final version of the paper.
>     - We randomly selected an anchor road, computed its representation similarity with other roads (according to cosine similarity), and displayed the top 10 most similar roads. The anchor road is red, the top 10 similar roads by the First Law are blue, and those by both laws are orange.
>     - With only the First Law, similar roads are much closer to the anchor. With both laws, similar roads are farther but have similar geographic configurations, as shown by comparing street view images. This demonstrates that the Third Law ensures similar representations for roads with similar configurations, regardless of distance.
>
> 3. For the GPU memory, the proposed method does not require more GPU memory as the number of roads (nodes) grows. Because we use a technique called subsampling (refer to **Sample** in Fig. 1 and the text description from line 207). In particular, for each iteration (forward and backward propagation), we only sample a subgraph with a fixed size of nodes (4000 nodes in our setting). We observe that the GPU memory usage is less than 10GB, and thus the proposed method can run on a single RTX 3090 GPU. In the implementation of graph learning (in a single graph), if the graph is not very large, we usually load the graph into the memory [1]. If the graph is extremely large and cannot be loaded into the memory, we can first sample the index of the nodes, and then read related edges and node features from the disk.
>
> 4.  Thanks very much for raising this interesting question. In current literature on road network representation, the changes in road network itself have not been considered, because its changes are relatively slow, and usually require several months or years. We also follow their settings. Actually, graph neural networks can be trained on certain graphs and tested on other similar graphs, and maintain competitive results. (For example, the PPI datasets are several graphs of proteins, and learned GNN models are tested on unseen proteins to predict the label of nodes. [2,3]) Thus, if there are only some small changes, we do not need to re-train the model, we just need to update the input data and get the output representation with the same model. If there are significant changes in the road network, we can try to re-train the model, and re-train is not very time-consuming. We just need several hours to train the model on Tesla V100 GPU, which is even slower than on a single RTX 3090 GPU.
>
> Response to Limitations:
>
> 1. (Refer to "Response to Weaknesses", 2)
> 2. (Refer to "Response to Questions", 4)
>
> ---
>
> [1] https://docs.dgl.ai/generated/dgl.data.DGLDataset.html#dgl.data.DGLDataset
>
> [2] Inductive Representation Learning on Large Graphs. In NeurIPS.
>
> [3] Simple and Deep Graph Convolutional Networks. In ICML.

---

> > ### Comment · Reviewer_ggbs · 2024-08-13
> >
> > Thanks for your response. I keep my original score.

---

> ### Author Response · Authors · 2024-08-10
> **Additional Questions**
>
> Dear Reviewer ggbs,
>
> Thanks very much for providing the constructive and motivating feedback! Can you please let us know whether we have addressed all your questions and whether you have any additional feedback?
>
> Thank you!

---

### Official Review · Reviewer_UWD1 · 2024-07-13

**Soundness:** 4
**Presentation:** 3
**Contribution:** 3
**Rating:** 6
**Confidence:** 4

**Summary:**

This paper introduces Garner, dual geographic-configuration-aware graph contrastive learning framework for road representation learning. Both the third law of geography and the first law of geography are considered by using street view images, geographic configuration aware graph augmentation,  and spectral negative sampling. The effectiveness of Garner is tested on three downstream tasks:  road function classification, road traffic prediction, and visual road retrieval.

**Strengths:**

1. The contributions of the proposal Garner framework are clearly highlighted.
2. The idea of jointly considering the third law of geography and the first law of geography is intriguing.
3. The statistical significance of the first two tasks are listed which is very good.

**Weaknesses:**

1. IMHO, the emphasis of learning the third law of geography is a bit oversold. Why can the averaging pool of CLIP-image embeddings of SVI along a road represent its geographic configuration? To me, the SVI pooled embedding can be simply treated as the visual feature of each road.
2. In terms of Section 4.2, other than the norm-based measure as the similarity measure, can you do an ablation study on different similarity measures?
3. Some math symbols are not well-explained which makes the paper a bit hard to read. For example, what are L_S, L_K, D_S, D_K? L and D are defined, but the subscriptions need to be explained. What are \mathbf{x}_i and \mathbf{x}_j in Equation 7?

**Questions:**

1. Why can the averaging pool of CLIP-image embeddings of SVI along a road represent its geographic configuration?
2. Could you do an ablation study on different similarity measures in Section 4.2?
3. Under Equation 8, you wrote "we design the negative sample based on Z and K, ..., by discriminating positive samples from negatives". Could you explain the intuition why the discriminative between positive and negative samples can minimize Equation 8?

**Limitations:**

The author need to add a paragraph about their limitations and some potential negative societal impact of their work

---

> ### Author Rebuttal · Authors · 2024-08-07
>
> The authors thank the reviewer for providing constructive and detailed comments, which may significantly improve this paper. We are pleased to inform you that all your concerns have been successfully addressed. Please see the detailed response to each of your comments listed below.
>
>
> Response to the weaknesses:
>
> 1. In A-Xing Zhu's paper [1] on the Third Law of Geography, the term "geographic configuration" is defined as "the makeup and the structure of geographic variables over some spatial **neighborhood** around a point." In our scenario, the target variable is the output representation of roads, and thus we should include variables and features that can effectively describe the roads. The street view images (SVIs), which provide the visual features of a road, also include important descriptions of a road. For example, the surface condition of a road and the number of lanes in a road (, which is not available for the majority of OSM data,).  Also, the geographic configuration includes descriptions of a point itself and its neighborhood. Street view images describe both the visual features of a road and its **neighborhood**, e.g., its surrounding buildings, regions etc. Therefore, we consider SVIs a suitable proxy to describe the geographic configuration of a road. To use SVIs in our model, we need to: (1) encode the images into vectorized representations; (2) aggregate multiple street view images along each road segment. To this goal, our implementation is "the averaging pool of CLIP-image embeddings of SVI along a road."
> 2. Following your suggestion, we have conducted an ablation study with different similarity measures. As mentioned in the paper, popular choices for similarity measure includes: norm/distance-based similarity, cosine similarity, Gaussian kernel. However, Gaussian kernel $\exp ( -\frac{(a - b)^2}{2 \sigma^2} )$ is a monotonic function of distance, and also a bijective function of distance. Thus we only conduct experiments on norm/distance-based similarity and cosine similarity. The results are listed in the following table. We observe that, with different similarity metrics, the model achieves very **similar** results.
>
>    **Road Function Prediction on Singapore**
>
>    | Metric | Micro-F1 (%) $\uparrow$ | Macro-F1 (%) $\uparrow$ | AUROC (%) $\uparrow$ |
>    | ------ | ----------------------- | ----------------------- | -------------------- |
>    | norm   | 81.40 $\pm$ 0.30        | 62.45 $\pm$ 0.64        | 93.27 $\pm$ 0.22     |
>    | cosine | 81.30 $\pm$ 0.34        | 62.26 $\pm$ 0.56        | 92.94 $\pm$ 0.21     |
>
>    **Road Function Prediction on NYC**
>
>    | Metric | Micro-F1 (%) $\uparrow$ | Macro-F1 (%) $\uparrow$ | AUROC (%) $\uparrow$ |
>    | ------ | ----------------------- | ----------------------- | -------------------- |
>    | norm   | 82.97 $\pm$ 0.16        | 47.22 $\pm$ 0.42        | 89.30 $\pm$ 0.21     |
>    | cosine | 82.95 $\pm$ 0.18        | 46.98 $\pm$ 0.56        | 89.13 $\pm$ 0.18     |
>
>    **Road Traffic Inference on Singapore**
>
>    | Metric | MAE $\uparrow$  | RMSE $\uparrow$ | MAPE $\uparrow$   |
>    | ------ | --------------- | --------------- | ----------------- |
>    | norm   | 2.80 $\pm$ 0.03 | 3.52 $\pm$ 0.04 | 0.579 $\pm$ 0.030 |
>    | cosine | 2.82 $\pm$ 0.02 | 3.54 $\pm$ 0.03 | 0.585 $\pm$ 0.024 |
>
>    **Road Traffic Inference on NYC**
>
>    | Metric | MAE $\uparrow$  | RMSE $\uparrow$ | MAPE $\uparrow$   |
>    | ------ | --------------- | --------------- | ----------------- |
>    | norm   | 3.30 $\pm$ 0.02 | 4.40 $\pm$ 0.03 | 0.207 $\pm$ 0.002 |
>    | cosine | 3.32 $\pm$ 0.02 | 4.44 $\pm$ 0.03 | 0.208 $\pm$ 0.002 |
>
> 3. Sorry for the confusion. Following the conventions in graph learning, the subscript of $\boldsymbol{L}\_{\boldsymbol{S}}$ indicates that this is a graph Laplacian matrix induced by some adjacency matrix $\boldsymbol{S}$, which is similar to $\boldsymbol{D}$, the degree matrix. In particular, $\boldsymbol{D}\_{\boldsymbol{S}}$ is the degree matrix of a graph with adjacency matrix $\boldsymbol{S}$, i.e., $(\boldsymbol{D}\_{\boldsymbol{S}})\_{i, i} := \sum_{i=1}^{n} \boldsymbol{S}\_{i}$, and $\boldsymbol{L}\_{\boldsymbol{S}} := \boldsymbol{D}\_{\boldsymbol{S}} - \boldsymbol{S}$. $\boldsymbol{D}\_{\mathcal{K}}$ and $\boldsymbol{L}\_{\mathcal{K}}$ are the degree matrix and graph Laplacian matrix of the complete graph $\mathcal{K}$ respectively. Equation 7 is listed as follows:
>    $$
>    usc\_{\mathcal{G}} = \min\_{\boldsymbol{x} \in \\{0, 1\\}^n - \\{\boldsymbol{0}, \boldsymbol{1}\\}} \frac{\sum_{(i, j) \in \mathcal{E}} (\boldsymbol{x}\_i - \boldsymbol{x}\_j)^2}{\sum_{(i, j)}(\boldsymbol{x}\_i - \boldsymbol{x}\_j)^2}
>        = \min\_{\boldsymbol{x} \in \\{0, 1\\}^n - \\{\boldsymbol{0}, \boldsymbol{1}\\}} \frac{\boldsymbol{x}^T \boldsymbol{L}\_{\mathcal{G}} \boldsymbol{x}}{ \boldsymbol{x}^{T} \boldsymbol{L}\_{\mathcal{K}} \boldsymbol{x}},
>    $$
>    where $\boldsymbol{x}$ is a n-dimensional vector, $\boldsymbol{x}\_i$ is the $i$-th element of vector $\boldsymbol{x}$. $\forall i$, $\boldsymbol{x}\_i$ is either $0$ or 1, but $\boldsymbol{x}$ cannot be a vector of all $0$s or $1$s. These explanations are represented as $\boldsymbol{x} \in \\{0, 1\\}^n - \\{\boldsymbol{0}, \boldsymbol{1}\\}$. The semantic meaning of this equation is that, the sparsest cut problem partitions nodes into two subsets and $\\{0, 1\\}$ denotes the which subset a node should belong to.

---

> > ### Comment · Reviewer_UWD1 · 2024-08-10
> > **I will keep my score**
> >
> > I thank the authors for their further explanation and I would like to keep my score.

---

> ### Author Response · Authors · 2024-08-07
> **[Rebuttal by Authors - 2]**
>
> Response to questions:
>
> 1. (Please refer to response to weakness 1.)
> 2. (Please refer to response to weakness 2.)
> 3. Sorry for the confusions in the draft. Our goal in Section 4.5 is to **design** the negative sample $\bar{\mathcal{G}}^{[1]}$ according to Equation 8 for the mutual information estimator in Equation 4 & 5, where the positive sample is $\mathcal{G}^{[1]}$. We get some inspirations to design the negative sample from minimizing the sparsest cut problem (Equation 8), but not to strictly minimize Equation 8. In particular, we find that in the sparsest cut problem
>    $$
>    \min\_{\boldsymbol{Z} \in \mathbb{R}^{n \times f}} {\frac{\operatorname{tr} (\boldsymbol{Z}^{T} \boldsymbol{L}\_{\boldsymbol{S}} \boldsymbol{Z})}{\operatorname{tr} (\boldsymbol{Z}^{T} \boldsymbol{L}\_{\mathcal{K}} \boldsymbol{Z})}},
>    $$
>    where the complete graph $\mathcal{K}$ could be a very good negative sample if we are going to minimize the numerator. Recall that in Section 4.3, leveraging SGC (simple graph convolution) as the graph encoder can minimize the numerator, and thus the complete graph $\mathcal{K}$ is a very good negative sample. Maximizing the denominator $\operatorname{tr} (\boldsymbol{Z}^{T} \boldsymbol{L}\_{\mathcal{K}} \boldsymbol{Z})$ is non-trivial, while minimizing the denominator is easy to achieve, through another SGC encoder. Minimizing the denominator is opposite to what we want, so we use it as the negative sample.
>
>
>
> Response to Limitations:
>
> Thanks for your suggestion. We would like to include the following two paragraphs in the final version of the paper.
>
> This paper is based on the First Law of Geography and the Third Law of Geography. Though the two laws are generally true, the method in this paper may fail where the two laws are not applicable. For example, the First Law may fail in extremely large areas or with limited data [1]. Also, we assume that the street view images are good proxies to describe the geographic configurations of roads.
>
> The potential negative societal impact includes: (1) Our method requires street view images (SVIs) along roads. However, SVIs may not be updated and thus our methods may provide outdated information. Also, SVIs cannot provide everyday changes in a city; (2) our method currently does not consider adversarial attacks from data, and thus may provide incorrect information for downstream tasks if it is attacked.
>
> ---
>
> [1] Spatial prediction based on Third Law of Geography. Annals of GIS, 2018.

---

> ### Author Response · Authors · 2024-08-10
> **Additional Questions**
>
> Dear Reviewer UWD1,
>
> Thanks very much for providing the constructive and motivating feedback! Can you please let us know whether we have addressed all your questions and whether you have any additional feedback?
>
> Thank you!

---

### Official Review · Reviewer_MjEv · 2024-07-15

**Soundness:** 2
**Presentation:** 3
**Contribution:** 2
**Rating:** 4
**Confidence:** 3

**Summary:**

This paper proposes a novel framework, termed Garner, for road network representation learning that leverages both the Third Law of Geography and the First Law of Geography. It emphasizes both spatial proximity and geographic configuration similarity.  Street view images  are used to capture similarities in road surroundings by geographic configuration-aware graph augmentation, spectral negative sampling, and a dual contrastive learning objective. ​Experiments demonstrate that integrating the Third Law significantly improves performance in downstream tasks like road function prediction, traffic inference, and visual road retrieval.

**Strengths:**

S1: Street view images from OpenStreetMap are used as proxies for geographic configurations.

S2: Considering the reverse implication of the Third Law: roads with dissimilar configurations should have dissimilar representations.

S3: The proposed method is validated on Singapore and New York City road networks with street view images.

**Weaknesses:**

W1: The novelty of this paper is limited, as geographic configuration has been modeled in many studies on urban computing.

W2: The choice of the lose, ie, to maximize the MI between the original graph $G^{[0]}$ and the augmented graph $G^{[1]}$, is not well justified and is difficult for me to understand.

W3: Several key factors, such as points of interest (POIs), are not taken into consideration.

**Questions:**

Q1: Why maximize the MI between the original graph $G^{[0]}$ and the augmented graph $G^{[1]}$? The two graphs have different semantics.

**Limitations:**

Adequately addressed

---

> ### Author Rebuttal · Authors · 2024-08-07
>
> The authors would like to thank the reviewer for providing constructive comments. We have made the following clarifications to address your concerns.
>
> Response to the Weaknesses:
>
> 1. The novelty of our paper is listed as follows:
>    - We introduce the **Third Law of Geography**, a fundamental principle in geographical sciences, which has not been explored in previous studies in road network representation learning.
>    - We design **novel graph contrastive learning** techniques, i.e., **geographic configuration-aware graph augmentation** and **spectral negative sampling**, where spectral negative sampling is inspired by the sparsest cut problem and spectral graph sparsification in spectral graph theory.
>    - According to [1], the definition of "geographic configuration" is "the makeup and the structure of geographic variables over some spatial **neighborhood** around a point." Thus, the term "geographic configuration" requires information from both a spatial entity and its **neighborhood**.
>
>    To the best of our knowledge, all of the above have not been explored in the literature of geospatial entity representation learning. We hope this clarification makes our novelty much easier to follow.
>
> 2. The original graph $\mathcal{G}^{[0]}$ and the augmented graph $\mathcal{G}^{[1]}$ contain different information from different views. Our goal is to fuse information from both views and generate better output representations of road segments. In particular, the augmented graph $\mathcal{G}^{[1]}$ is constructed according to the Third Law of Geography. As written in our paper, "Previous studies show that information from different views can be fused properly by maximizing their mutual information (MI) [2,3], which can also improve the quality of representations [3]." Inspired by these, we maximize the MI between the original graph $\mathcal{G}^{[0]}$ and the augmented graph $\mathcal{G}^{[1]}$ so as to allow the output representation to learn from the view constructed according to the Third Law of Geography.
>
> 3. Thanks for your suggestions. We do not include POIs in the paper due to the following reasons:
>    - **Focus on the Third Law of Geography**: Our paper focuses on the Third Law of Geography. In our paper, street view images (SVIs) can already model geographic configurations and the third law very well. Thus, SVIs are sufficient to illustrate our concepts and achieve our research objectives.
>    - **Distribution and Coverage of POIs**: We would like to argue that POIs are currently not appropriate for our problem. A major issue is observed that POIs are unevenly distributed in cities [10], and a large portion of road segments do not have nearby POIs. In contrast, SVIs cover almost every road in a city, and we can sample SVIs for each road segment.
>    - **Commercial Bias of POIs**: Another issue is that POIs are provided by Internet Web Maps for map search, and thus their contents are mainly for commercial functions (e.g., restaurants), leading to poor performance in non-commercial regions such as industrial or residential areas [10, 11]. However, there are many roads outside commercial regions where POIs may not provide sufficient information for subsequent analytical tasks. In contrast, SVIs can provide meaningful information across diverse regions.
>    - **Future Work**: We acknowledge that it is very interesting to identify an appropriate way to consider POIs in Garner in future work. And we will try our best to achieve this in the future.
>
>
>
> ---
>
> [1] Spatial prediction based on Third Law of Geography. Annals of GIS, 2018.
>
> [2] Learning representations by maximizing mutual information across views. In NeurIPS.
>
> [3] Contrastive multi-view representation learning on graphs. In ICML.
>
> [4] Graph convolutional networks for road networks. In SIGSpatial 2019.
>
> [5] On representation learning for road networks. ACM Trans. Intell. Syst. Technol 2021.
>
> [6] Spatial structure-aware road network embedding via graph contrastive learning. In EDBT 2023.
>
> [7] Robust road network representation learning: When traffic patterns meet traveling semantics. In CIKM 2021
>
> [8] Jointly contrastive representation learning on road network and trajectory. In CIKM 2022
>
> [9] Learning effective road network representation with hierarchical graph neural networks. In KDD 2020
>
> [10] Urban2vec: Incorporating street view imagery and pois for multi-modal urban neighborhood embedding. In AAAI 2020
>
> [11] Knowledge-infused Contrastive Learning for Urban Imagery-based Socioeconomic Prediction. In WWW 2023

---

> > ### Comment · Reviewer_MjEv · 2024-08-10
> > **Thanks for your reply**
> >
> > I agree with you on the use of POIs, street view images are sufficient for the illustration of your idea.
> > However, it's not entirely reasonable for me to say that maximizing MI aims at fusion. Maximizing MI retains only the shared information between the two graphs. In other words, the geographic information you expected may be lost during this.

---

> > > ### Author Response · Authors · 2024-08-11
> > > **Thanks for your reply**
> > >
> > > Dear Reviewer MjEv,
> > >
> > > Thank you for your thoughtful feedback and for allowing us the opportunity to address your concerns. We are pleased to have resolved some of the issues you raised and appreciate your constructive comments.
> > >
> > > We agree with your opinion that maximizing mutual information (MI) is appropriate for fusing views that share information. We would like to provide further clarification on the fusion of the two graphs in our approach:
> > >
> > > 1. **Shared Information Between Graphs**: Both graphs, $\mathcal{G}^{[0]}$ and $\mathcal{G}^{[1]}$, have shared information of geographic configurations as the same node features. As described in Section 4.1, we prepare their node features as concatenation of projected road features and projected street view image embeddings. Specifically, the node features in both graphs are $\boldsymbol{H}^{[0]} = \text{concat}([\boldsymbol{C} \boldsymbol{W}_c, \boldsymbol{X} \boldsymbol{W}_x])$, where $\boldsymbol{C}$ is the matrix of geographic configurations, $\boldsymbol{X}$ is the matrix of road features (from map data) and $\boldsymbol{W}_c$ and $\boldsymbol{W}_x$ are learnable projection matrix. This "joint representation" is a widely used method in multi-modality and multi-view fusion [1].
> > > 2. **Differences between Graphs**: The only difference between the original graph $\mathcal{G}^{[0]}$ and the augmented graph $\mathcal{G}^{[1]}$ are their edges, which provide different views of underlying road segments. Specifically, edges in $\mathcal{G}^{[0]}$ describe the connectivity of roads in the map, while edges in $\mathcal{G}^{[1]}$ connect roads with very similar geographic configurations. By maximizing the mutual information between them, the model can learn some "high-level factors whose influence spans multiple views" [2], for example, the similarity of road segments considering the third law. Our ablation study also demonstrates that, building the augmented graph $\mathcal{G}^{[1]}$ for contrastive learning significantly enhances the learning performance of our proposed method.
> > >
> > > Thank you again for your valuable feedback.
> > >
> > >
> > > [1] Multimodal Machine Learning: A Survey and Taxonomy. TPAMI.
> > >
> > > [2] Learning Representations by Maximizing Mutual Information Across Views. In NeurIPS 2019.

---

> > > > ### Comment · Reviewer_MjEv · 2024-08-13
> > > > **Thanks for your reply**
> > > >
> > > > Your reply does not address my concerns. A suitable loss function is the core of a learning model.
> > > >
> > > > In some studies[1], models learn not only share information but also exclusive information via mutual information.
> > > >
> > > > [1] Sanchez, Eduardo Hugo, Mathieu Serrurier, and Mathias Ortner. "Learning disentangled representations via mutual information estimation." Computer Vision–ECCV 2020: 16th European Conference, Glasgow, UK, August 23–28, 2020, Proceedings, Part XXII 16. Springer International Publishing, 2020.

---

> ### Author Response · Authors · 2024-08-10
> **Additional Questions**
>
> Dear Reviewer MjEv,
>
> Thanks very much for providing the constructive and motivating feedback! Can you please let us know whether we have addressed all your questions and whether you have any additional feedback?
>
> Thank you!

---

> > ### Comment · Reviewer_MjEv · 2024-08-13
> >
> > I raised my score to  4: Borderline reject since my concerns are partially addressed.

---

> > > ### Author Response · Authors · 2024-08-13
> > >
> > > Dear Reviewer MjEv,
> > >
> > > Thanks for your reply. We also agree that careful designs are required to keep both shared information and exclusive information. And we have achieved this in our method. We would like to provide both clarifications and experimental results as follows.
> > >
> > > 1. As mentioned in [1], models can learn both shared information and exclusive information via mutual information by considering both global and local mutual information.
> > > 2. The loss in our method is one of the most widely used MI estimators,  Jensen-Shannon MI estimator [2, 3], which considers the global-local mutual information by maximizing the mutual information between a patch of an image (local high-level representations) and the image itself (high-level "global" representation).
> > > 3. In the field of **graph contrastive learning**, generating an augmented graph and using mutual information maximization to **fuse** different views of graphs, keep both shared information and exclusive information, and enhance the representation are widely adopted [4, 5, 6]. Our designs are also based on this literature to fuse different views. Besides, the experimental results in 4-6 also indicate the effectiveness of the fusion and keeping both shared information and exclusive information (i.e., geographic configurations).
> > > 4. In our **ablation study**, line 2 and line 3 in Table 5 show that the maximizing the mutual information between the original graph $\mathcal{G}^{[0]}$ and the augmented graph $\mathcal{G}^{[1]}$ to fuse different views and keep exclusive information (i.e., geographic configurations) can significantly improve the learning performance. In particular, we archive up to 25% improvement.
> > > 5. We have conducted a downstream task named **visual road retrieval** (Table 4 in the paper), aiming at finding which road a street view image should belong to. In this experiment, our method, which uses SVIs as part of the input and models the Third Law of Geography, performs significantly better than the baselines, which do not consider geographic configurations. The experimental results indicate that the representation learned a lot from the geographic configurations (exclusive information).
> > > 6. We have conducted a **case study**, which is included in the attached PDF under "Author Rebuttal by Authors". In the case study, we randomly selected an anchor road, computed its representation similarity with other roads and displayed the top 10 most similar roads. By considering the third law in our method, similar roads are farther but have similar geographic configurations. This also reveals that our method can keep exclusive information of geographic configurations.
> > >
> > >
> > > ---
> > >
> > > [1] Learning disentangled representations via mutual information estimation. In ECCV.
> > >
> > > [2] Learning deep representations by mutual information estimation and maximization. In ICLR.
> > >
> > > [3] Deep Graph Infomax. In ICLR.
> > >
> > > [4] Contrastive Multi-View Representation Learning on Graphs. In ICML 2020.
> > >
> > > [5] Graph Contrastive Learning with Adaptive Augmentation. In WWW 2021.
> > >
> > > [6] Multi-Scale Contrastive Siamese Networks for Self-Supervised Graph
> > > Representation Learning. In IJCAI.

---

### Author Rebuttal · Authors · 2024-08-07

We would like to express our sincere gratitude to all the reviewers for their careful review and constructive comments. Here, we briefly summarize our response to all the comments.
1. A case study can be found in the uploaded PDF file.
2. More ablation studies and concrete examples have been provided to demonstrate the critical role of the Third law of Geography in road network representation learning and clarify the motivations of the proposed approach.
3. As suggested by the reviewers, more clarifications on the modules of the proposed methods have been provided to make the proposed method easy to understand. In addition, sensitivity tests and more ablation studies have been conducted to show the peculiarities of the proposed approach.
4. Experimental settings are clarified to make the experiments in this work convincing.
5. As suggested by the reviewers, we have conducted discussions to distinguish the proposed approach from other related works, thus enhancing the novelty and contributions of the paper.
6. How we will improve the paper according to all the reviewers' suggestions has been illustrated.

Again, we thank all the reviewers, and we will carefully revise the paper following all the suggestions raised.

---

### Decision · Program_Chairs · 2024-09-25

**Decision:**

Accept (poster)

**Comment:**

This paper introduces a novel framework, Garner, which leverages graph contrastive learning to integrate the Third Law of Geography into road network representation learning. The method is technically sound, and the authors provide comprehensive empirical validation, showing significant improvements across several downstream tasks. While the paper is strong in novelty and effectiveness, it would benefit from clearer explanations of the Third Law's role, additional discussion of spectral negative sampling, and improved presentation. Despite these minor issues, the paper's contributions are significant, and I recommend acceptance with revisions for clarity.